# Concentration and source changes of HONO during the COVID-19 lockdown in Beijing

Yusheng Zhang[1], Feixue Zheng[1], Zemin Feng[1,2], Chaofan Lian[3,4], Weigang Wang[3,4*], Xiaolong Fan[1,5,6], Wei Ma[1], Zhuohui Lin[1], Chang Li[1], Gen Zhang[7], Chao Yan[8,9], Ying Zhang[1,9], Veli-Matti Kerminen[1,8] , Federico Bianch[1,8], Tuukka Petäjä[1,8], Juha Kangasluoma[1,8], and Markku Kulmala[1,8], Yongchun Liu[1*]

1. Aerosol and Haze Laboratory, Advanced Innovation Center for Soft Matter Science and Engineering, Beijing University of Chemical Technology, Beijing, 100029, China

2. College of Chemical Engineering, North China University of Science and Technology, Tangshan 063210, China

3. State Key Laboratory for Structural Chemistry of Unstable and Stable Species, Beijing National Laboratory for Molecular Sciences (BNLMS), CAS Research/Education Center for Excellence in Molecular Sciences, Institute of Chemistry, Chinese Academy of Sciences, Beijing 100190, China

4. University of Chinese Academy of Sciences, Beijing 100049, China

5. Center for Excellence in Regional Atmospheric Environment, Institute of Urban Environment, Chinese Academy of Sciences, Xiamen 361021, China

6. Fujian Key Laboratory of Atmospheric Ozone Pollution Prevention, Institute of Urban Environment, Chinese Academy of Sciences, Xiamen 361021, China

7. State Key Laboratory of Severe Weather & Key Laboratory of Atmospheric Chemistry of China Meteorological Administration (CMA), Chinese Academy of Meteorological Sciences (CAMS), Beijing 100081, China

8. Institute for Atmospheric and Earth System Research/Physics, Faculty of Science, University of Helsinki, P.O. Box 64, FI-00014, Finland

9. Joint International Research Laboratory of Atmospheric and Earth System Science, School of Atmospheric Sciences, Nanjing University, Nanjing, China.

*Correspondence: Yongchun Liu (liuyc@buct.edu.cn) and Weigang Wang (wangwg@iccas.ac.cn)

**Abstract:**

Nitrous acid (HONO) is an important precursor of OH radicals which affects not only the sinks of primary air pollutants but also the formation of secondary air pollutants, whereas its source closure in the atmosphere is still controversial due to a lack of experiment validation. In this study, the HONO budget in Beijing has been analyzed and validated through the coronavirus disease (COVID-19) lockdown event, which resulted in a significant reduction in air pollutant emissions, providing a rare opportunity to understand the HONO budget in the atmosphere. We measured HONO and related pollutants from January 1, 2020, to March 6, 2020, which covered the Chinese New Year (CNY) and the COVID-19 lockdown. The average concentration of HONO decreased from $0.97 \pm 0.74$ ppb before CNY to $0.53 \pm 0.44$ ppb during the COVID-19 lockdown, accompanied by a sharp drop of $NO_x$ and the greatest drop of NO (around 87%). HONO budget analysis suggests that vehicle emissions were the most important source of HONO during the nighttime ($53 \pm 17\%$) before CNY, well supported by the decline of their contribution to HONO during the COVID-19 lockdown. We found that the heterogeneous conversion of $NO_2$ on ground surfaces was an important nighttime source of HONO ($31 \pm 5\%$), while that on aerosol surfaces was a minor source ($2 \pm 1\%$). Nitrate photolysis became the most important daytime source during the COVID-19 lockdown compared with that before CNY, resulting from the combined effect of the increase in nitrate and the decrease in NO. Our results indicate that reducing vehicle emissions should be an effective measure for alleviating HONO in Beijing.

## 1. Introduction

As the most vital oxidant in the troposphere, OH radicals not only govern the sink of most trace compounds but also affect the production of secondary pollutants by initiating photochemical reactions in the atmosphere. Nitrous acid (HONO) is an important primary precursor of OH radicals (Kulmala and Petäjä, 2011; Zhang et al., 2023c). Photolysis of HONO can contribute 60% (Tan et al., 2018) and sometimes even 92% (Xue et al., 2020) to OH production in the morning. Therefore, HONO can indirectly promote the formation of both secondary aerosols (Zhang et al., 2019b) and ozone (Zhang et al., 2022a). In addition, HONO can react with histamine to form carcinogens, such as nitrosamines, after entering the human body (Farren et al., 2015). Thus, understanding the sources of HONO in the atmosphere has been a hot topic for several decades, but it is still far from closed (Jiang et al., 2022). Intensive studies have been carried out on HONO measurements and source analysis (Liu et al., 2020c; Liu et al., 2020d; Zheng et al., 2020; Zhang et al., 2020; Xue et al., 2020; Zhang et al., 2019a; Liu et al., 2019b). The concentrations of HONO in the atmosphere range from a few ppt in remote areas (Spataro et al., 2016) to several ppb, even several tens ppb in heavily polluted areas (Liu et al., 2019b; Liu et al., 2020c; Liu et al., 2020d; Zheng et al., 2020).

The sources of atmospheric HONO consist of direct emissions and secondary formation in the atmosphere. Direct emissions include soils, biomass burning, vehicles, indoor air, and livestock farming. Soil emissions, which depend on soil types, microorganisms, water content, temperature, and pH (Kulmala and Petäjä, 2011; Weber et al., 2015; Kim and Or, 2019), are important sources of HONO. Biomass burning, often occurs in the summer and autumn when wheat/corn is harvested and wildfires are common (Zhang et al., 2019b; Sun et al., 2017; Sun

et al., 2018; Peng et al., 2020). Vehicle emissions are considered an important source of HONO
in traffic-intensive areas (Kramer et al., 2020; Li et al., 2021). This source is more important at
nighttime compared with daytime (Zhang et al., 2016; Fu et al., 2019; Liu et al., 2020d).
Recently, indoor emissions have also been proposed as a potential HONO source (Xue, 2022),
which is related to the ventilation from high HONO concentrations in indoor air to low HONO
concentrations in outdoor air (Zhang et al., 2019b). Livestock farming is a previously
overlooked source of HONO, especially in agricultural areas.
Secondary formation of HONO includes gas-phase reaction between NO and OH radicals,
photolysis of particulate nitrate, and heterogeneous reaction of $NO_2$ on ground and particulate
matter surfaces, including photochemical heterogeneous reaction of $NO_2$. Gas phase reaction
between NO and OH, photolysis of nitrate particles, and light-enhanced conversion of $NO_2$ are
the main daytime sources of HONO (Liu et al., 2019c; Liu et al., 2020d; Zhang et al., 2022b).
Furthermore, acid replacement processes may be a non-negligible source of daytime HONO in
locations affected by soil-borne mineral dust deposition (Vandenboer et al., 2014). The
heterogeneous reaction of $NO_2$ on various surfaces is widely regarded as an important source
of HONO (Han et al., 2016; Liu et al., 2020b).
Table S1 summarizes the sources of HONO at various locations. The type of observation
site often has a great impact on the source intensity and contribution proportion of each source
of HONO. In natural ecological areas or Antarctic stations with little human activity, the
photolysis of nitrate is the main source of HONO during the day, and its contribution is much
higher than the homogeneous reaction of NO and OH (Bond et al., 2023; Tang et al., 2024). In
the ocean or areas close to the sea, the heterogeneous transformation of $NO_2$ becomes the main

source of HONO, and the transformation on the aerosol surface may be more important than that on the ground (Xing et al., 2023). In smoke collected near wildfires, it was found that the heterogeneous conversion contribution of $NO_2$ can reach 85%, making it the most important source of HONO (Chai et al., 2021). Emissions from soil and biological soil crusts are important in some areas where vegetation and soil are exposed (Meusel et al., 2018). For three different types of observation sites: rural, suburban, and urban, the relative importance of sources is also obviously different. In rural areas, there are usually no traffic activities, and are mainly affected by agricultural activities and animal husbandry, so traffic emissions can be ignored. During periods of intensive agricultural activity, soil emissions are the main source of HONO, accounting for up to 80% (Liu et al., 2019c), When there is little agricultural activity, the reaction of NO and OH and the heterogeneous transformation of $NO_2$ on the ground become the two main sources in rural areas (Xue et al., 2020; Song et al., 2022), accounting for up to 70%. In rural areas with developed animal husbandry, its direct emissions can contribute 39-45% of HONO (Zhang et al., 2023a). Suburbs are mostly covered by vegetation, with a small number of villages nearby. The heterogeneous conversion of $NO_2$ is the main source of HONO, which can reach 70% of HONO sources (Fu et al., 2019; Ye et al., 2023). For highways, tunnels, and urban areas with heavy traffic, traffic emissions usually dominate HONO sources, accounting for 40% to 80% of HONO sources (Xu et al., 2015; Zhang et al., 2019c; Liu et al., 2020d; Kramer et al., 2020). In some ordinary urban areas where traffic activities are not so intensive, the heterogeneous conversion of $NO_2$ and the reaction of NO and OH are also the main sources of HONO in addition to traffic sources. It can be seen that the relative importance of different sources is often affected by the type of emission source near the observation site.

Although intensive studies have been performed on HONO sources, the contributions of different sources are still controversial (Zhou et al., 2011; Liu et al., 2014; Wu et al., 2019; Kramer et al., 2020; Meng et al., 2020). For the same type of observation area, the contribution of each source still diverges in different studies. For example, in mixed residential, commercial, and traffic areas, the importance of traffic emissions varies greatly. In some studies, it accounts for as much as 50% (Liu et al., 2020d; Zhang et al., 2019a; Tong et al., 2016), while in some studies, it can be ignored (Zhang et al., 2020). A similar situation exists for the heterogeneous conversion of $NO_2$. Some studies suggest that this process is not important (Tong et al., 2015; Zhang et al., 2019c; Zhang et al., 2022b), while some studies believe that it can contribute at least 70% of HONO (Meng et al., 2020; Zhang et al., 2020; Jia et al., 2020). It should be noted that the contribution of $NO_2$ heterogeneous reaction to HONO greatly depends on the choice of $NO_2$ uptake coefficient ($\gamma_{NO2}$), which varies from $10^{-8}$ to $10^{-4}$ in different studies (Meng et al., 2020; Liu et al., 2020b; Ge et al., 2019; Liu et al., 2015; Liu et al., 2020d). Vehicle emissions also have similar characteristics because the HONO emission rate strongly depends on the emission factor, i.e. the ratio of $HONO/NO_x$ (Kramer et al., 2020; R. Kurtenbach et al., 2001; Zhang et al., 2019c), which ranges from 0.03% to 2.1% (Liao et al., 2021). For other HONO sources, the relative importance is affected by many parameters, such as reaction kinetics for photolysis of nitrate, OH concentrations for homogeneous reaction between NO and OH, emission fluxes for soil emissions, and so on. Thus, the HONO budget still has a large uncertainty. In particular, it is an open question how to prove the importance of a specific reaction pathway or a source of atmospheric HONO.

Special events taking place on large spatial scales provide us with an alternative

opportunity to disclose the mysteries of the HONO budget because of obvious and potentially
large changes in some of the HONO sources. During the Spring Festival in 2020, the lockdown
measures during the new coronavirus disease -19 (COVID-19) pandemic led to a significant
reduction in primary emissions from traffic and industries. The magnitude and speed of changes
in air pollutant emissions have been considered the largest changes in the history of modern
atmospheric chemistry (Kroll et al., 2020). We conducted continuous field observations of
HONO and other air pollutants from January 1, 2020, to March 6, 2020, in downtown Beijing,
aimed at understanding the changes in HONO concentrations and sources during the lockdown
period compared to that before.
**2. Experimental section**
**2.1 Field measurements.**
Observations were carried out at the Aerosol and Haze Laboratory, Beijing University of
Chemical Technology (AHL/BUCT), which has been described in our previous work (Liu et
al., 2020d). Briefly, it is located on the west campus of BUCT, around 550 m from the west
third-ring road of Beijing, which is a typical urban observation site. The station is on the rooftop
of a 5-story building (about 18 m from the ground). HONO was measured with a homemade
Water-based Long-Path Absorption Photometer (LOPAP, Institute of Chemistry, Chinese
Academy of Sciences), which has been deployed in field observation studies (Tong et al., 2016;
Chen et al., 2020) and has been proven to be a stable and credible instrument for HONO
measurements (Crilley et al., 2019). The principle of this instrument is similar to that of a
commercial LOPAP (QUMA). Briefly, gas-phase HONO absorbed by deionized water ($\geq$ 18.2
M$\Omega$) in a stripping coil reacts with N-(1-naphthyl) ethylenediamine-dihydrochloric acid (0.077
mmol $L^{-1}$) in an acidic solution (2 mmol $L^{-1}$ sulfanilamide in 0.12 mol $L^{-1}$ HCl) to form an azo
dye, which is measured at 550 nm with a spectrometer equipped with a LWCC (Liquid
Waveguide Capillary Cell, LWCC-3250, WPI, USA). The sampling rate was 1 L $min^{-1}$
controlled by a flow meter and a diaphragm pump. The flow rate of absorption liquid was 0.5
ml $min^{-1}$ controlled by a peristaltic pump. The limit of detection of the LOPAP was 0.01 ppb
for a sampling duration of 60 s. The instrument was calibrated with nitrite standard solution
before and after each measurement about every three weeks and calibrated by zero air every
24 hours to check zero drift. An overestimation of HONO concentration (6.7%), calibrated in
control experiments with 100 ppb of $NO_2$ at 50% RH due to the interference of $NO_2$ in the
sampling inlet (about 30 cm of Teflon tube), was accounted for when we calculating the HONO
concentrations in this work.
A set of commercial analyzers for $NO_x$, $SO_2$, CO, and $O_3$ (Thermo Scientific 42i, 43i, 48i,
49i) were also available. Notably, the $NO_2$ measured by 42i includes HONO, and we have
corrected it. $PM_{2.5}$ was measured using a Tapered Element Oscillating Microbalance (TEOM,
Thermo Fisher Scientific, 1405). The chemical composition of non-refractory $PM_{2.5}$ (NR-$PM_{2.5}$)
was measured using a Time-of-flight Aerosol Chemical Speciation Monitor (ToF-ACSM,
Aerodyne). Meteorological parameters including temperature, RH, pressure, wind speed and
direction, and ultraviolet radiation (A and B) were measured using a weather station (AWS 310
at AHL/BUCT station, Vaisala). The planetary boundary layer (PBL) height and visibility were
measured using a ceilometer (CL51, Vaisala) and a visibility sensor (PWD22, Vaisala),
respectively. The photolysis rate ($J_{NO2}$) was measured via a continuous measurement of the
actinic flux in the wavelength range of 285-375 nm using a $J_{NO2}$ filter-radiometer (2-pi-$J_{NO2}$
radiometer, Metcon). All instruments used in the measurement as well as their detection limits
are shown in Table S2.
**2.2 HONO budget calculation.**
Potential sources of HONO include direct emissions (vehicle emissions, soil emissions, indoor
emissions, biomass combustion), the gas-phase reaction between NO and OH radicals, the
photolysis of nitrate in particulate matter, and the heterogeneous reaction of $NO_2$ on the ground
and particulate matter surfaces. The sources including vehicle emissions ($E_{vehicle}$), soil
emissions ($E_{soil}$), the reaction of NO and OH ($P_{NO-OH}$), the photolysis of particulate nitrate
($P_{nitrate}$), and the heterogeneous reaction of $NO_2$ ($P_{aerosol}$ and $P_{ground}$). At present, there are
relatively few studies on indoor emissions. Biomass combustion is an unimportant HONO
source in downtown Beijing in winter according to a previous study (Zhang et al., 2019b). Thus,
these two sources are not accounted for in this work. The major sinks of HONO, including dry
deposition ($L_{deposition}$), the homogeneous reaction with OH radicals ($L_{HONO-OH}$), photolysis
($L_{photolysis}$), and vertical and horizontal transport ($T_{trans}$), are considered.

The calculation method and details in parameterization are shown in Table 1. Briefly, the

budget and estimated concentration of HONO can be calculated according to the following
equations,
$$\frac{dHONO}{dt} = E_{soil} + E_{vehicle} + P_{NO-OH} + P_{nitrate} + P_{aerosol} + P_{ground} -$$
$$L_{photolysis} - L_{HONO-OH} - L_{deposition} - T_{trans} \tag{1}$$
$$HONO_{est,t_2} = HONO_{obs,t_1} + Sources_{t_2} - Sinks_{t_2} \tag{2}$$
where $\frac{dHONO}{dt}$ is the change rate of HONO mixing ratios (ppb h$^{-1}$), $HONO_{est,t_2}$ is the
estimated concentration of HONO at time t$_2$, while $HONO_{obs,t_1}$ is the observed concentration
of HONO at time $t_1$. Given that the result of potential source contribution function (PSCF, Fig
S2), the source distribution of HONO between BCNY and COVID was highly similar and the
trend of HONO was similar (Pearson'r=0.78) between BUCT and Institute of Atmospheric
Physics (IAP, 8 km away from BUCT), the steady state analysis on HONO is appliable and
reasonable even though the lifetime of HONO is several minutes in the atmosphere. In addition,
the instrumentation time resolution of LOPAP was 6 s. We calculated the variation coefficient
for the datasets with different time resolutions, i.e., 1 h $vs$ 6 s. A small variation coefficient of
~0.02-0.05 implies that a small uncertainty of the HONO budget might result from the lifetime
of HONO. Thus, we think the possible uncertainty should not have a large influence on our
conclusions when the budget is compared at a fixed site between two different periods. The
input parameters for the parameterization scheme are detailed in Table S3 (M0).
The emission rate ($E_{HONO}$, ppb h$^{-1}$) of soil and vehicle were calculated based on the
emission flux ($F_{HONO}$, g m$^{-2}$ s$^{-1}$), the PBL height ($H$, m), and the conversion factor ($\alpha$, g m$^{-3}$ s$^{-1}$
to ppb h$^{-1}$). For vehicle emissions, according to our previous research at the same site, the
emission factor (EF, HONO/NO$_x$) was selected as 1.09% (Liu et al., 2020d), which is
comparable to the actual values in Hong Kong (1.2 ± 0.4% and 1.24 ± 0.35%) (Liang et al.,
2017; Xu et al., 2015), Guangzhou (1.0%) (Li et al., 2012), Beijing (1.3% and 1.41%) (Zhang
et al., 2019c; Meng et al., 2020), and other places. For secondary formation, the calculation of
the production rate ($P_{HONO}$, ppb h$^{-1}$) is shown in Table 1, in which k$_1$ is the rate constant of the
quasi-first order reaction (s$^{-1}$). For the heterogeneous reaction of NO$_2$, we calculated the
conversion rate in the light of Eqs. (3)-(5):

$$k_{het}^0 = \frac{HONOcorr,t_2 - HONOcorr,t_1}{\overline{NO_2} \times (t_2 - t_1)}$$     (3)

$$k_{het}^{co} = \frac{2 \times \left[ \frac{HONOcorr,t_2}{CO_{t_2}} \times \overline{CO} - \frac{HONOcorr,t_1}{CO_{t_1}} \times \overline{CO} \right]}{(t_2 - t_1) \times \left[ \frac{NO_{2,t_2}}{CO_{t_2}} + \frac{NO_{2,t_1}}{CO_{t_1}} \right] \times \overline{CO}}$$

$$= \frac{2 \times \left[ \frac{HONOcorr,t_2}{CO_{t_2}} - \frac{HONOcorr,t_1}{CO_{t_1}} \right]}{(t_2 - t_1) \times \left[ \frac{NO_{2,t_2}}{CO_{t_2}} + \frac{NO_{2,t_1}}{CO_{t_1}} \right]} \qquad (4)$$

$$k_{het} = \frac{1}{2} \times (k_{het}^0 + k_{het}^{co}) \qquad (5)$$

where $k_{het}$ is the quasi-first-order rate constant of the transformation to HONO (s$^{-1}$), $k_{het}^0$
and $k_{het}^{co}$ are the reaction rate constants after uncalibrated and CO calibrated, respectively
(Zhang et al., 2020). To decrease the contribution of boundary layer height variation on the $k_{het}$
calculations, we normalized HONO concentration to CO concentration as the same as reported
in the literature (Zhang et al., 2019c; Li et al., 2012). $\overline{NO_2}$ and $\overline{CO}$ are the mean
concentration of NO$_2$ and CO from t$_1$ to t$_2$. $CO_t$ and $NO_{2,t}$ are mixing ratios of CO and NO$_2$,
respectively, at the measuring time t. HONO$_{corr,t}$ (ppb) is the HONO concentration corrected
after subtracting the primary emissions (including vehicle and soil emissions, and the HONO
produced by the homogeneous reaction of NO and OH and the photolysis of nitrate) at the
measuring time t according to Eq. (6):
$$HONO_{corr,t} = HONO_t - E_{soil,t} - E_{vehicle,t} - P_{NO-OH,t} - P_{nitrate,t} \quad (6)$$

it is worth noting that the $HONOcorr$ only accounted for vehicle exhausts in previous HONO
budget studies. This may overestimate the contribution of heterogeneous reactions to HONO
sources because other emission sources and homogeneous reactions should also contribute to
HONO.
Meanwhile, when estimating the upper limit of the contribution of heterogeneous
reactions, we take a small conversion factor (HONO/NOx) of 0.4% as the lower limit of vehicle
emissions, in contrast to the normal value of 1.09% (Liu et al., 2020d). We normalize the
$EI_{NO_X}$ caused by the vehicle with the measured $NO_X$ during the observations. This method
has also been widely used in previous studies (Liu et al., 2019b; Li et al., 2018). In addition,
soil emissions are calculated using the lower limit (Oswald et al., 2013). The mean value of
$k_{het}$ during the BCNY (before the Chinese New Year) was 0.0051 h$^{-1}$, while it was 0.006 h$^{-1}$
in the COVID-19 lockdown, which are consistent with previous studies, such as Ji'nan (0.0068
h$^{-1}$) (Li et al., 2018) and Shanghai (0.007 h$^{-1}$) (Wang et al., 2013), while less than those in
Shijiazhuang (0.016 h$^{-1}$) (Liu et al., 2020c), Kathmandu (0.014 h$^{-1}$) (Yu et al., 2009), and
Guangzhou (0.016 h$^{-1}$) (Qin et al., 2009).

Table 1. Summary of parameters for HONO sources and sinks

| HONO formation/loss pathways | Calculations | Parameters | Reference |
|---|---|---|---|
| Soil emissions → HONO | | $F_{HONO,soil}$ | 1 |
| Vehicle emissions → HONO | $E_{HONO} = \alpha \times F_{HONO}/H$ | $F_{HONO,vehicle} = (EI_{NOx,vehicle}/A) \times (HONO/NOx)_{vehicle}$ | 2 |
| NO + OH → HONO | | $k_{NO\text{-}OH} = 7.2 \times 10^{-12}$ cm$^3$ molecule$^{-1}$ s$^{-1}$ | 3 |
| $NO_3^- \xrightarrow{h\nu} HONO$ | | $J_{NO3^-} = 8.24 \times 10^{-5}/3.59 \times 10^{-7} \times J_{HNO3,MCM}$ | |
| $NO_2 + H_2O \xrightarrow{aerosol\ surface} HONO$ | $P_{HONO} = 3600 \times k \times c_{precursor}$ | $k_{het} = (\gamma_{NO2} \times A_s \times \omega/4) \times Y_{HONO}$ | 4 |
| $NO_2 + H_2O \xrightarrow{ground\ surface} HONO$ | | $k_{het} = (\gamma_{NO2} \times \delta \times \omega/4H) \times Y_{HONO}$ | |
| $HONO \xrightarrow{h\nu} NO + OH$ | $L_{photolysis} = 3600 \times J_{HONO} \times HONO$ | $J_{HONO,\ MCM}$ | |
| $HONO + OH → H_2O + NO_2$ | $L_{HONO\text{-}OH} = 3600 \times k_{HONO\text{-}OH} \times HONO \times OH$ | $k_{HONO\text{-}OH} = 6 \times 10^{-12}$ cm$^3$ molecule$^{-1}$ s$^{-1}$ | 5 |
| HONO deposition | $L_{deposition} = (3600 \times V_d \times HONO)/H$ | $V_d = 0.001$ m s$^{-1}$ | 6 |
| HONO transport (vertical and horizontal) | $T_{trans} = k_{dilution} \times (HONO\text{-}HONO_{background})$ | $k_{dilution} = 0.23$ h$^{-1}$ | 7 |

$F_{HONO,soil}$ (soil emission flux) was calculated by the temperature-dependent HONO emission flux on grasslands with a water content of 35% to 45%. A is the urban area of Beijing, $EI_{NOx,vehicle}$ is the emission inventory of NO$_x$ from vehicle (g s$^{-1}$). The calculation of the HONO emission flux, during BCNY, was based on the hourly NO$_x$ emission inventory of Beijing vehicles ($F_{HONO}=F_{NOx} \times (HONO/NOx)_{vehicle}$), while during COVID-19, it was combined with the hourly average traffic index (www.nitrafficindex.com). The $(HONO/NOx)_{vehicle}$ was selected as 1.09% (Liu et al., 2020d). $c_{precursor}$ is the concentration of the precursor (ppb). The OH concentration was estimated using the same method as in the previous study (Liu et al., 2020c). The mean photolysis frequency of nitrate ($J_{NO3^-}$) was normalized to the measured UV light intensity. $A_s$ is the surface area concentration of the reaction surface (m$^2$ m$^{-3}$); $\omega$ is the average molecular velocity (m s$^{-1}$); $\gamma$ is the uptake coefficient of the precursor, was assumed to be $2 \times 10^{-6}$; $Y_{HONO}$ is the yield of HONO. δ is the surface roughness, in this study, we used 3.85 for our calculation (Liu et al., 2020d). $HONO$ and $HONO_{background}$ are the HONO concentrations at the observation site and background site, respectively. $J_{HONO}$ is simulated in a box model using $J_{NO2}$ data observed at our site.

1: (Oswald et al., 2013). 2: (Yang et al., 2019). 3: (Liu et al., 2020c). 4: (Liu et al., 2020d). 5: (Kanaya et al., 2007). 6: (Han et al., 2017b). 7: (Dillon et al., 2002).

We further derived the uptake coefficient of $NO_2$ ($\gamma_{NO2}$) on both ground and particle
surfaces according to Eq. (7).

$$k_{het} = \frac{\gamma_{NO2} \times A_s \times \omega}{4} \times Y_{HONO} \tag{7}$$

The calculated $\gamma_{NO2}$ ranged from $1 \times 10^{-6}$ to $3 \times 10^{-6}$. Therefore, we choose
$2 \times 10^{-6}$ to calculate the heterogeneous yield of HONO, which is comparable with those
derived in urban environments like Ji'nan ($1.4 \times 10^{-6}$) (Li et al., 2018) and the laboratory
experiments ($10^{-7}$ to $10^{-6}$) (Han et al., 2013; Stemmler, 2007; Han et al., 2017a) on different
particles, but lower than the uptake coefficient of $10^{-5}$ reported in other studies (Zhang et al.,
2020; Ge et al., 2019).
The OH concentration was calculated according to Eq. (8), which is based on the function
of the photolysis rates ($J$) of $O_3$ and $NO_2$, and the $NO_2$ mixing ratio ($NO_2$).

$$OH = \frac{4.1 \times 10^9 \times (J_{NO2})^{0.19} \times (J_{O1D})^{0.83} \times (140 NO_2 + 1)}{0.41 NO_2^2 + 1.7 NO_2 + 1} \tag{8}$$

Notably, this parameterization scheme was developed based on measurements at rural sites
(Ehhalt and Rohrer, 2000), where NOx concentrations were lower than in urban environments.
Alicke et al. (Alicke, 2002) found that OH concentrations estimated with this scheme were in
good agreement with those calculated according to a pseudo-steady state method during the
pollution period in urban environments (such as Milan), although some uncertainty was
expected. In our previous study (Liu et al., 2020d), we also found that the estimated OH
concentrations using this method were comparable with those observed values in the North
China Plain (Tan et al., 2019). Thus, daytime OH concentrations estimated using this method
should be overall credible although the uncertainty is inevitable. The nocturnal OH
concentration in North China generally varied from $1.0 \times 10^5$ molecules $cm^{-3}$ (Ma et al., 2019;
Tan et al., 2018) in winter to $5\times10^5$ molecules cm$^{-3}$ in summer (Tan et al., 2017; Tan et al.,
2020). We further parameterized the nocturnal OH concentrations according to atmospheric
temperature to reflect the seasonal variations of OH concentration. Fig. S3 summarizes the
observed OH concentrations in the North China Plain. The results estimated in this study are
slightly lower than those observed in Wangdu (Rural), but almost consistent with those in
Beijing (Urban) and Huairou (Suburb). In summary, we should be optimistic about the
estimation of OH concentration. Then a sensitivity analysis was performed to understand the
influence of the uncertainty of OH concentration on HONO sources as discussed in Section 3.3.

The loss rate of HONO, including dry deposition ($L_{deposition}$), homogeneous reaction

with OH radicals ($L_{HONO-OH}$), photolysis ($L_{photolysis}$), and vertical and horizontal transport
($T_{trans}$), were calculated using the equations shown in Table 1. Where $J_{HONO}$ is the photolysis
rate of HONO (s$^{-1}$), $k_{HONO-OH}$ is the second-order reaction rate constant between HONO and
OH, $V_d$ is the dry deposition rate of HONO, and $K_{dilution}$ is the dilution rate (including both
vertical and horizontal transport). The details are described in our previous work (Liu et al.,
2020c; Liu et al., 2020d).

Oracle Crystal Ball (version 11.1.2.4, Oracle's software for modeling, prediction,

simulation, and optimization) (Rahmani et al., 2023) to evaluate the overall uncertainty of the
parameterization through Monte Carlo simulations. The details are shown in Text S2 in the SI.
**3. Results and discussion**
**3.1 Air quality during observations.**
Figure 1 shows the time series of the concentration and relative proportion of non-refractory
components in PM$_{2.5}$, trace gases (SO$_2$, O$_3$, CO, NO, NO$_2$, and NO$_x$), and meteorological

parameters (temperature, relative humidity (RH), and pressure). We divide the sampling period

into two sub-periods, i.e., P1 from January 1 to January 24 (BCNY, before the Chinese New

Year) and P2 from January 25 to March 6 (COVID-19 lockdown).

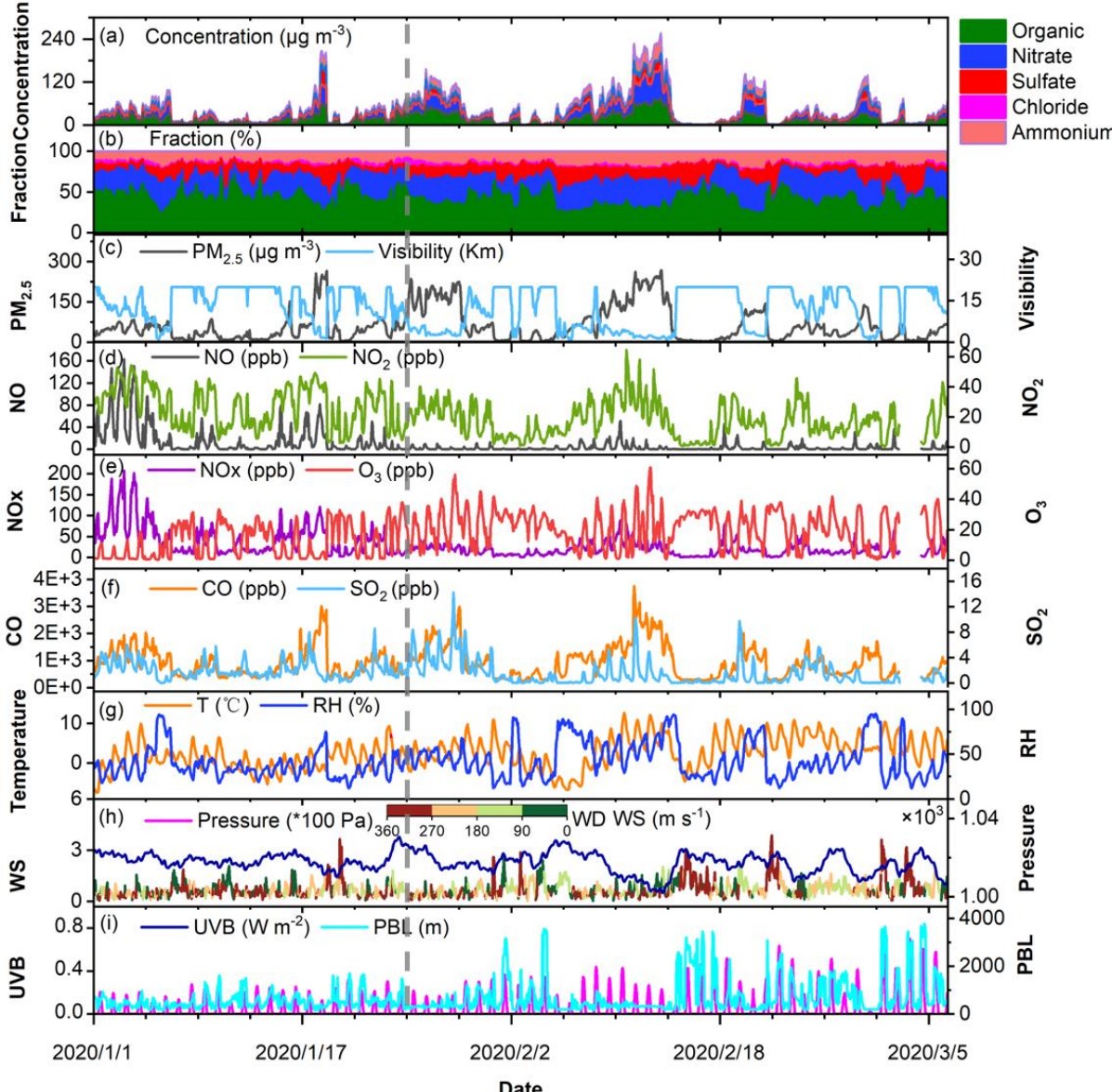

**Figure 1.** An overview of the measurement of the mass concentrations of the different

components of non-refractory-PM2.5 (NR-PM2.5), the mass fraction of the individual

components, PM2.5, and meteorological parameters, NOx (NO, NO2), O3, CO and SO2 in 1-hour

average from 1 January to 6 March 2020. Meteorological parameters consist of visibility, PBL

heights, UVB, wind speed, wind direction, Pressure, RH, and temperature. The observations

are divided into two phases (P1:2020.01.01-2020.01.24 and P2:2020.01.25-2020.03.06).

It can be seen from Fig. 1 that during P1, there was only one heavy pollution incident
lasting one to two days, while there were two serious pollution events lasting more than two
days ($PM_{2.5} > 75$ μg m$^{-3}$) in the P2 stage. Table S4 summarizes the statistical results of the wind
speed, $PM_{2.5}$, RH, T, HONO, trace gases, and NR-$PM_{2.5}$ for the entire measurement period.
During P1, the measured concentration of $PM_{2.5}$ varied between 0.2-288 μg m$^{-3}$ and the mean
concentration was $47.2 \pm 44.5$ (mean $\pm 1\sigma$) μg m$^{-3}$. In contrast, they were 0.3-258 μg m$^{-3}$ and
$69.9 \pm 67.2$ μg m$^{-3}$, respectively, during P2. The mean concentrations of $NO_x$ decreased
significantly ($P < 0.05$) from $45.35 \pm 38.86$ ppb in P1 to $19.44 \pm 14.42$ ppb in P2, dropping by
about 57%. This is close to the reduction amplitude (50%) reported by Wang et al. (Wang et
al., 2020a) but lower than that (76%) proposed by Lv et al (Lv et al., 2020). In particular, the
NO mean concentrations dropped from $18.42 \pm 29.24$ ppb (ranging from 0.03 to 163 ppb) in
P1 to $2.4 \pm 5.46$ ppb (ranging from 0.01 to 51 ppb). The average hourly concentration of $NO_2$
in the P1 phase was $26.9 \pm 13.4$ ppb, while it was $17.18 \pm 11.3$ ppb in P2. The $NO_2$
concentration dropped by about 36% from P1 to P2, which is similar to the recently reported
findings (ranging from 36% to 53%) (Zhao et al., 2020; Wang et al., 2020b; Wang et al., 2021).
According to the emission inventory of $NO_x$, traffic and industry contributed 46.7% and 31.3%
to $NO_x$ emissions in Beijing, respectively (Zheng et al., 2014). This means the decrease in $NO_x$
concentration should be explained by both reductions in traffic and industrial emissions (Lv et
al., 2020; Wang et al., 2020a; Zhao et al., 2020). In particular, traffic emissions during P2
should play an important role in local NO reduction. However, as the temperature and
ultraviolet light irradiation increased and the $NO_x$ concentration decreased (Kroll et al., 2020;
Le et al., 2020), the average concentration of $O_3$ during P2 was $21.31 \pm 11.73$ ppb, which was
significantly ($P < 0.05$) higher than $12.16 \pm 10.79$ ppb during P1. This result is similar to the
71.4% increase in $O_3$ in Shijiazhuang during the same period (Liu et al., 2020c). The
concentrations of $SO_2$ were in the range of 0.02-8.56 ppb with a mean value of $2.09 \pm 1.35$ ppb
in P1, while it varied from 0.01 to 14.23 ppb with the mean concentration of $1.49 \pm 1.99$ ppb
during P2, suggesting slightly decreased contribution of coal combustion during P2 (Fig. 1i).
This is similar to that reported by Cui et al (Cui et al., 2020) and Shen et al (Shen et al., 2021).
In addition, it can be seen from Fig. 1 that the change trends of $PM_{2.5}$ and CO are synchronized,
which also means that both primary emissions and secondary generation contribute to the
accumulation of $PM_{2.5}$ concentration (Liu et al., 2020c).

It is worth noting that changes in atmospheric pollutant concentrations are affected by

both emissions and meteorology. Especially, during the lockdown period, meteorological
conditions in Beijing were not conducive to the dispersion of pollutants, thus the impact of
meteorological conditions on the concentration of these pollutants needs to be assessed. We
use the random forest algorithm of machine learning to remove the influence of meteorology
from air quality time series data by a deweather method. The details are present in Text S1 in
the SI. The model performs well in predicting the concentrations of pollutants compared to the
observations in both the training and test datasets (Table S5). The concentrations and relative
changes of each pollutant after deweather are recorded in Table S6. The $PM_{2.5}$ concentration
after deweather increased significantly from $45.22\pm28.56$ in P1 to $67.92\pm57.97$ $\mu g\ m^{-3}$ in P2 at
a confidence level of 0.05, with an increase of 50.2%. The mean concentration of HONO was
$0.89\pm0.37$ ppb in P1, while it decreased to $0.51\pm0.25$ ppb in P2, with a drop of 42.7%; The
concentrations of NO and $NO_2$ significantly decreased from $15.44\pm18.40$ and $23.28\pm7.28$ ppb
in P1 to $3.24\pm2.05$ and $16.43\pm5.98$ ppb in P2, respectively, which decreased by 79.0% and 29.4%
respectively; $SO_2$ decreased from $2.27\pm0.69$ in P1 to $1.48\pm1.18$ ppb in P2, a decrease of
approximately 34.8%; CO increased from $823.60\pm318.92$ in P1 to $896\pm488.29$ ppb in P2 (an
increase of 8.79%) and $O_3$ increased from $16.98\pm5.62$ to $22.60\pm4.10$ ppb, an increase of about
33.1%, which was much lower than the change range of observed values (75.1%). As shown
in Table S6, meteorological conditions have a significant impact on $O_3$ concentration. The
impact was +39.6% and +6.2% in P1 and P2, respectively. The impact of deweather on NO in
the two periods was -16.2% and +32.8%, respectively. It was -13.8% and -4.8%, respectively,
for $NO_2$. However, the changes of other species in the two periods after deweather fluctuated
between 2.3% and 7.8%. This implies that meteorological conditions have an important impact
on the concentrations of NO and $O_3$, while meteorological factors have little impact on HONO,
$SO_2$, CO, and $PM_{2.5}$.

It can be seen from Figure 1 combined with Table S4 in SI. All the major components of

$PM_{2.5}$, including sulfate, nitrate, ammonium, chloride, and organic aerosol, increased obviously
in P2 compared to P1. Throughout the entire observation period, organic matter and nitrate
dominated the composition of $PM_{2.5}$. The proportion of nitrate in inorganic salts increased to
31.2% in P2, up from 28.1% in P1. Although the sulfate concentration increased, its proportion
within inorganic salts slightly decreased on haze days, going from 16.5% in P1 to 15.2% in P2.
Thus, the ratio of $NO_3^-$ to $SO_4^{2-}$ during pollution events increased significantly from 1.76 in P1
to 2.10 in P2 ($P < 0.05$). This is similar to previous findings reported by Sun (Sun et al., 2020).
These findings suggest that the decrease in anthropogenic emissions during the P2 period
resulted in a significant reduction (After the T-test, it is significant at a confidence level of 0.01.)
in gas precursors (Table S4), but it did not lead to a corresponding reduction in secondary
aerosol species during periods of pollution. This is supported by the increased potential
secondary aerosol formation under pollution conditions (Sun et al., 2020). For example, higher
values of the SOR (sulfur oxidation ratio, molar fraction of sulfate in total sulfur including
sulfate and $SO_2$) and NOR (nitrogen oxidation ratio, molar fraction of nitrate in total nitrogen,
including nitrate and $NO_2$), i.e., 0.63 and 0.34, were observed in P2 than those (0.48 and 0.14)
in P1. Under stagnant weather conditions (wind speed $< 2$ m s$^{-1}$), higher temperatures and RH
as shown in Table S4 might facilitate the conversion from precursors into particles (Liu et al.,
2020d). The above results indicate that the air pollution dominated by secondary formation is
much more serious in P2, which is supported by both the increased concentration and the
greater number of pollution days in P2 than in P1, even though primary emissions decreased
obviously.
**3.2 Influence of Chinese New Year and the COVID-2019 epidemic event on HONO**
**concentration in Beijing.**
Figure 2 displays the time series of the HONO concentration, the HONO/$NO_2$ ratio, and the
traffic index (www.nitrafficindex.com). In Fig. 2b, there is a significant decrease in the traffic
index ($P < 0.05$), indicating reduced traffic congestion during the COVID-19 lockdown (P2
period) compared to the P1 period. The HONO/$NO_2$ ratio is frequently used to indicate the
conversion of $NO_2$ to HONO through heterogeneous reactions (Sun et al., 2013). A higher
HONO/$NO_2$ indicates that the heterogeneous conversion process plays a more significant role
in HONO production. However, as depicted in Fig. 2b, both the traffic index and HONO exhibit
a similar decreasing trend, while the HONO/NO$_2$ ratio remains relatively stable. Notably, both
the traffic index and the NO concentration experienced a steep decline after January 24,
coinciding with a significant decrease ($P < 0.05$) in HONO concentration. Furthermore, as
shown in Fig. S4, there is a strong correlation between HONO and NO$_x$ in both P1 and P2.
However, HONO concentration does not track PM$_{2.5}$ concentration well. These results imply
that HONO might be more influenced by vehicle emissions than by heterogeneous reactions
on aerosol surfaces. This contrasts with prior studies that heterogeneous reactions on aerosol
surfaces are the primary source of HONO in pollution events in Beijing (Liu et al., 2014; Cui
et al., 2018; Meng et al., 2020).

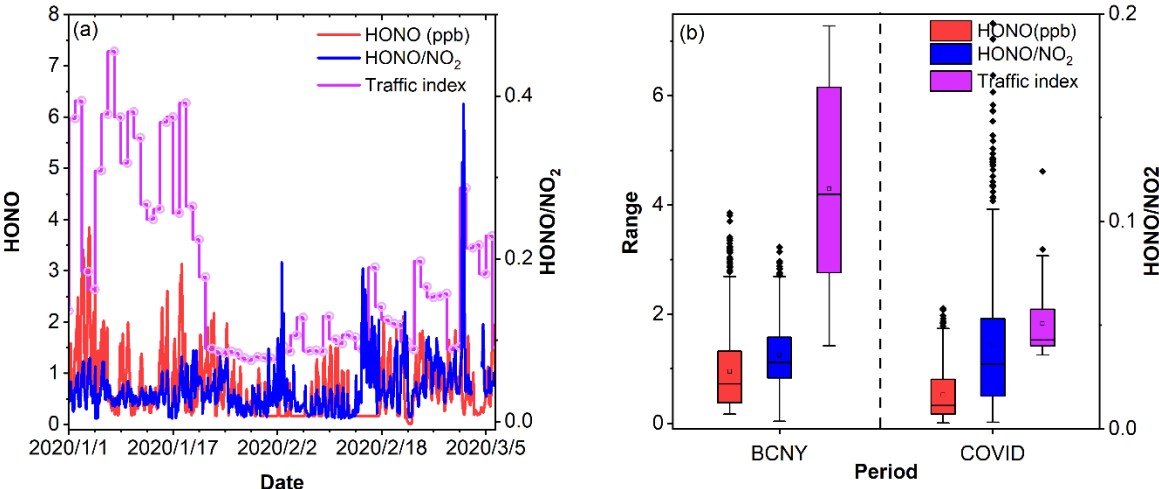


**Figure 2.** (**a**) Times series of HONO, traffic index, and HONO/NO$_2$, (**b**) Box plots of HONO,
HONO/NO$_2$, and the traffic index in Beijing during different periods (BCNY=P1, LOCK=P2).

Table S7 summarizes the mean concentrations of HONO, NO$_2$, NO, and PM$_{2.5}$ over the
two periods in this study as well as the data reported in previous studies. During P1, HONO
concentration ranged from 0.17 to 3.85 ppb, with a mean value of $0.97 \pm 0.74$ ppb. This
concentration is similar to previous observations, such as in Beijing, Xi'an, Jinan, Shanghai,
Hong Kong, and Rome, which all ranged from 0.95 to 1.15 ppb (Acker et al., 2006; Wang et
al., 2013; Xu et al., 2015; Huang et al., 2017; Liu et al., 2020d; Li et al., 2018). However, during
the COVID-19 lockdown, the HONO concentration decreased to $0.53 \pm 0.44$ ppb, representing
a drop of 45.3% compared with that in BCNY. After deweather, the HONO concentration
decreased significantly from $0.89 \pm 0.37$ in P1 to $0.51 \pm 0.25$ ppb in P2 at a confidence level of
0.05, with a decrease of 42.7%. This means that meteorology has little impact on HONO. This
value is comparable to the concentrations reported in the literature for clean days in December
2016 in Beijing ($0.5 \pm 0.2$ ppb) and in the winter of 2018 in Xiamen (0.52-0.61 ppb). At the
same time, as discussed in the previous section, the NO concentration decreased by nearly 87%
from BCNY to COVID-19 lockdown, and the $NO_2$ concentration dropped by about 36%.
Consequently, we can conclude that the concentrations of HONO, NO, and $NO_2$ were the most
affected pollutants during the COVID-19 lockdown period.
Figure 3 shows the diurnal curves of HONO, $NO_x$, NO, $NO_2$, $HONO/NO_2$, $O_3$, $SO_2$, and
$PM_{2.5} \times NO_2$ during P1 (BCNY) and P2 (COVID-19 lockdown). The black and red lines
represent P1 and P2, respectively. HONO shows a similar trend in both periods. After sunset,
HONO began to accumulate due to the attenuation of solar radiation and the development of
the boundary layer, reaching maximum values of $1.41 \pm 0.83$ ppb and $0.92 \pm 0.64$ ppb around
7:00 during P1 and P2, respectively. Subsequently, due to the impact of the boundary layer and
rapid photolysis, the HONO concentration gradually decreased and remained at a low level
until sunset, with the corresponding minimum value of $0.43 \pm 0.24$ ppb and $0.27 \pm 0.17$ ppb at
about 15:00. Similar to HONO, the $NO_2$ concentration shows an upward trend during the
morning rush hour. Its peak appeared at 7:00 (BCNY: $31.4 \pm 9.23$ ppb; COVID-19 lockdown:
$23.3 \pm 10.74$ ppb), and then dropped rapidly and remained at a low level due to photochemical
processes and the development of the boundary layer. The minimum concentration occurs
around 14:00 to 15:00 (BCNY: $18.17 \pm 10.69$ ppb; COVID-19 lockdown: $11.0 \pm 7.64$ ppb).
After sunset, $NO_2$ began to increase again. It is worth noting that during BCNY, both $NO_2$ and
NO exhibited a prominent evening peak, whereas there was no such evening peak during the
COVID-19 lockdown. Thus, $NO_x$ and $NO_2$ had similar changing trends, i.e., the morning peak
observed in both periods with the highest mean values of $65.93 \pm 50.37$ ppb and $31.7 \pm 21.47$
ppb in BCNY and COVID-19 lockdown, respectively.

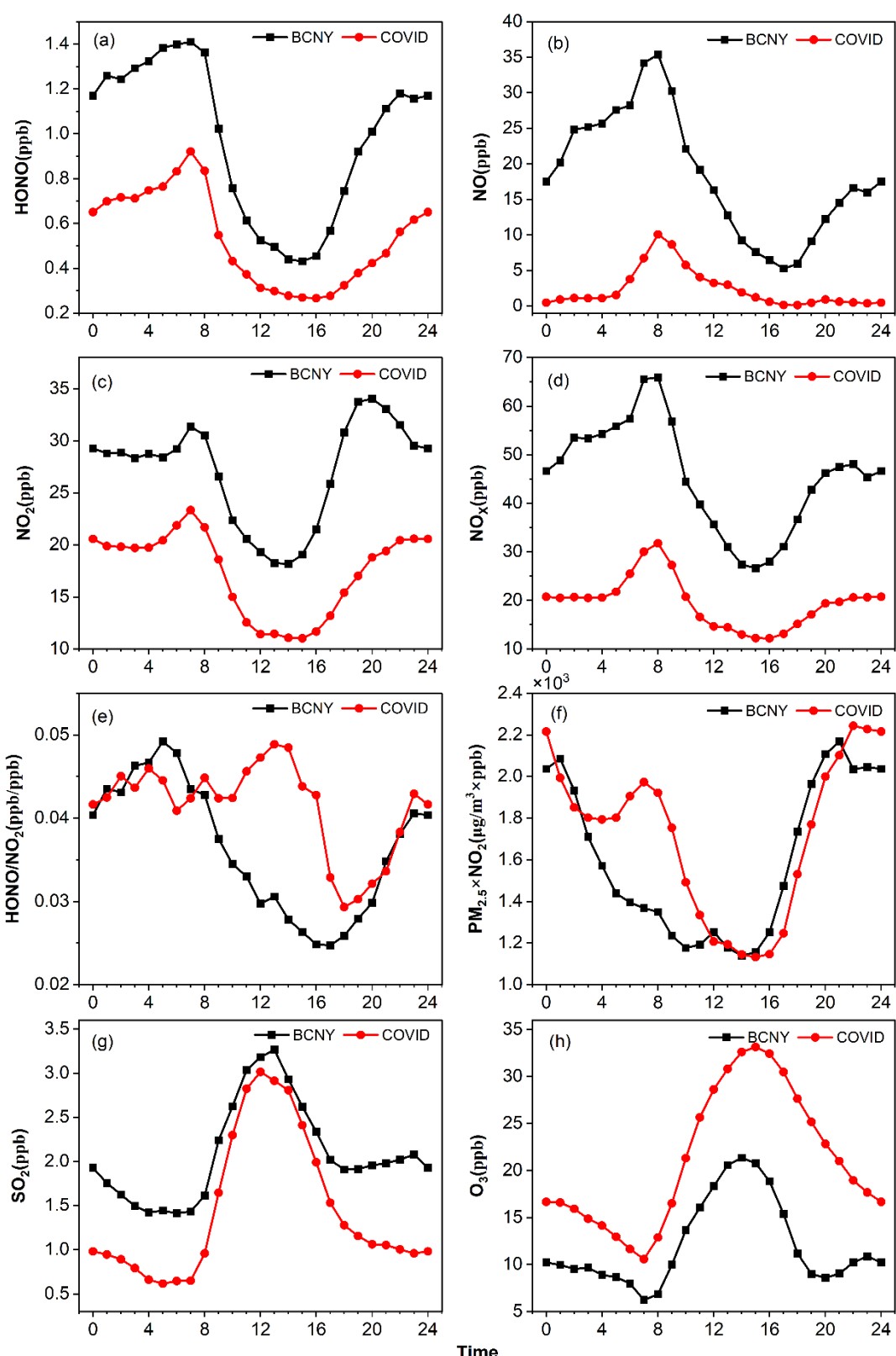


**Figure 3.** Diurnal variation of HONO, NO, $NO_2$, $NO_x$, $HONO/NO_2$, $PM_{2.5} \times NO_2$, $SO_2$, $O_3$. The

black lines are the diurnal curves before CNY and the red ones are during the COVID-19
lockdown.

NO and HONO showed a similar trend in P1. They began to decline continuously after
sunrise and continued to rise after sunset. The peaks of NO were $35.40 \pm 43.55$ ppb and $10.0 \pm$
12.67 ppb in P1 and P2, respectively. It is worth noting that the upward trend of NO
concentration in the afternoon of the P2 stage was not obvious, as the absolute concentration
of NO was very low. $O_3$ and HONO showed opposite diurnal curves, with the maximum $O_3$
concentrations occurring in the afternoon, which were $21.35 \pm 9.31$ ppb and $33.14 \pm 10.26$ ppb
in P1 and P2, respectively. $SO_2$ and $O_3$ exhibited similar trends, with the maximum values in
P1 and P2 were $3.26 \pm 2.19$ ppb and $3.01 \pm 3.06$ ppb at 13:00, and their lowest values were
$1.41 \pm 0.68$ ppb and $0.62 \pm 0.82$ ppb at 5:00 or 6:00.
Previous studies proposed that the heterogeneous reactions of $NO_2$ on the aerosol surface
play an important role in HONO production. Specifically, this pathway was considered the
major source of HONO on polluted days (Cui et al., 2018; Meng et al., 2020; Zhang et al.,
2020). $PM_{2.5} \times NO_2$ can be used as an indicator for the heterogeneous reaction of $NO_2$ on the
surface of aerosols (Cui et al., 2018). It was found that the value of $PM_{2.5} \times NO_2$ in P2 ($1697 \pm$
2142) was slightly higher than that in P1 ($1583 \pm 1967$). In the early morning, the product of
$PM_{2.5}$ and $NO_2$ in the P2 stage was even higher than that in the P1 stage. On the other hand, the
ratio of $HONO/NO_2$ is usually used to evaluate the formation of HONO during the conversion
of $NO_2$. As shown in Fig. 3, in the P1 stage, the $HONO/NO_2$ ratio shows a similar daily trend
to HONO, which began to rise after sunset and reached a peak at night and then decreased in
the early morning due to the increase of $NO_2$ concentrations and the photolysis of HONO. In
the P2 stage, the variation of $HONO/NO_2$ is different from that of the P1. The $HONO/NO_2$ in
the P2 period was higher than that in the P1 stage, especially in the daytime, although the values
of $HONO/NO_2$ in both stages (P1: $0.036 \pm 0.016$; P2: $0.041 \pm 0.038$) were lower than that
(0.052-0.080) reported by Cui et al (Cui et al., 2018). Subsequently, we further analyzed
$HONO_{corr}/NO_2$ (details shown in Sect. 2.2). The $HONO_{corr}/NO_2$ attributed to secondary
formation via heterogeneous reactions changed obviously after subtracting other secondary
HONO sources. As shown in Fig. S5, the daytime peak of $HONO_{corr}/NO_2$ in P2 became more
prominent compared with that in Fig. 3e, while the daytime (8:00 - 18:00) $HONO_{corr}/NO_2$
($0.022 \pm 0.014$) in P1 was significantly ($P < 0.05$) lower than that in P2 ($0.040 \pm 0.053$).
However, the HONO concentration decreased significantly as discussed above. These results
suggest that heterogeneous reactions of $NO_2$ on the aerosol surfaces may not be a major source
of HONO because the enhanced potential of heterogeneous reactions indicated by $PM_{2.5} \times NO_2$
and $HONO_{corr}/NO_2$ in P2 contrast with the decreased HONO concentrations compared to P1.
In summary, we propose that during our observation period, heterogeneous reactions of $NO_2$
should have a relatively minor contribution to HONO production.
**3.3 Relative change of different sources to HONO budget in Beijing during different**
**periods.**
Figure 4a-f shows the diurnal variation of HONO production or emission rates for these sources
at different stages, and Fig. 4g-l shows the budget of the HONO sources and sinks during P1
(BCNY) and P2 (COVID-19 lockdown). The HONO production rate via homogeneous
reaction between NO and OH in the P1 period was much higher than that in the P2 period,
especially during the daytime. The average rate decreased from $0.145 \pm 0.189$ ppb h$^{-1}$ in the P1
stage to 0.047 ± 0.073 ppb h$^{-1}$ in the P2 stage. The OH concentrations increased slightly from
P1 ($4.1×10^5 ± 5.8×10^5$ cm$^{-3}$) to P2 ($6.7×10^5 ± 1.0×10^6$ cm$^{-3}$). Therefore, the observed decrease
in HONO production rate via homogeneous reaction between NO and OH should be ascribed
to the substantial reduction of NO concentration as discussed above. It can be seen that the
homogeneous reaction between NO and OH is indeed an important source of HONO at night.
In previous studies, the nocturnal production of HONO via NO and OH was often ignored
because low nighttime OH concentrations were estimated (Fu et al., 2019). However, some
studies have shown that the observed nighttime OH concentrations in the Beijing urban area
can also be maintained in the order of $10^5$ molecules cm$^{-3}$ in winter, which also means that the
contribution of the reaction channels of NO and OH to HONO cannot be ignored. In the P1
stage, the homogeneous reaction between NO and OH accounted for 13 ± 5% of the nighttime
HONO sources. However, in the daytime, the homogeneous reaction between NO and OH was
the most important source of HONO, which accounted for up to 51 ± 32% of the daytime
HONO source. This is consistent with previous studies in urban Beijing (Gu et al., 2021; Jia et
al., 2020; Liu et al., 2021). Interestingly, a recent study proposed a new mechanism through
smog chamber experiments, that is, NOx photooxidation (reaction of NO and adsorbed HNO$_3$)
may be an important daytime HONO source (Song et al., 2023), although it has not yet been
verified by field observations. In the P2 stage, its proportion in the night was negligible due to
the dramatic decrease in NO concentration during the pandemic event, and the maximal
proportion of HONO sources in the daytime was also reduced to 25 ± 14%. It is worth noting
that the parameterization of OH concentration will introduce uncertainty to HONO sources.
Table S3 shows the sensitivity test for the HONO simulation. An increase of 10% and 200% in
OH concentration in M3 and M4 results in a 24-26% change in the HONO source. It means
that the accuracy of the OH measurement is important for understanding the source-sink
balance of the HONO.

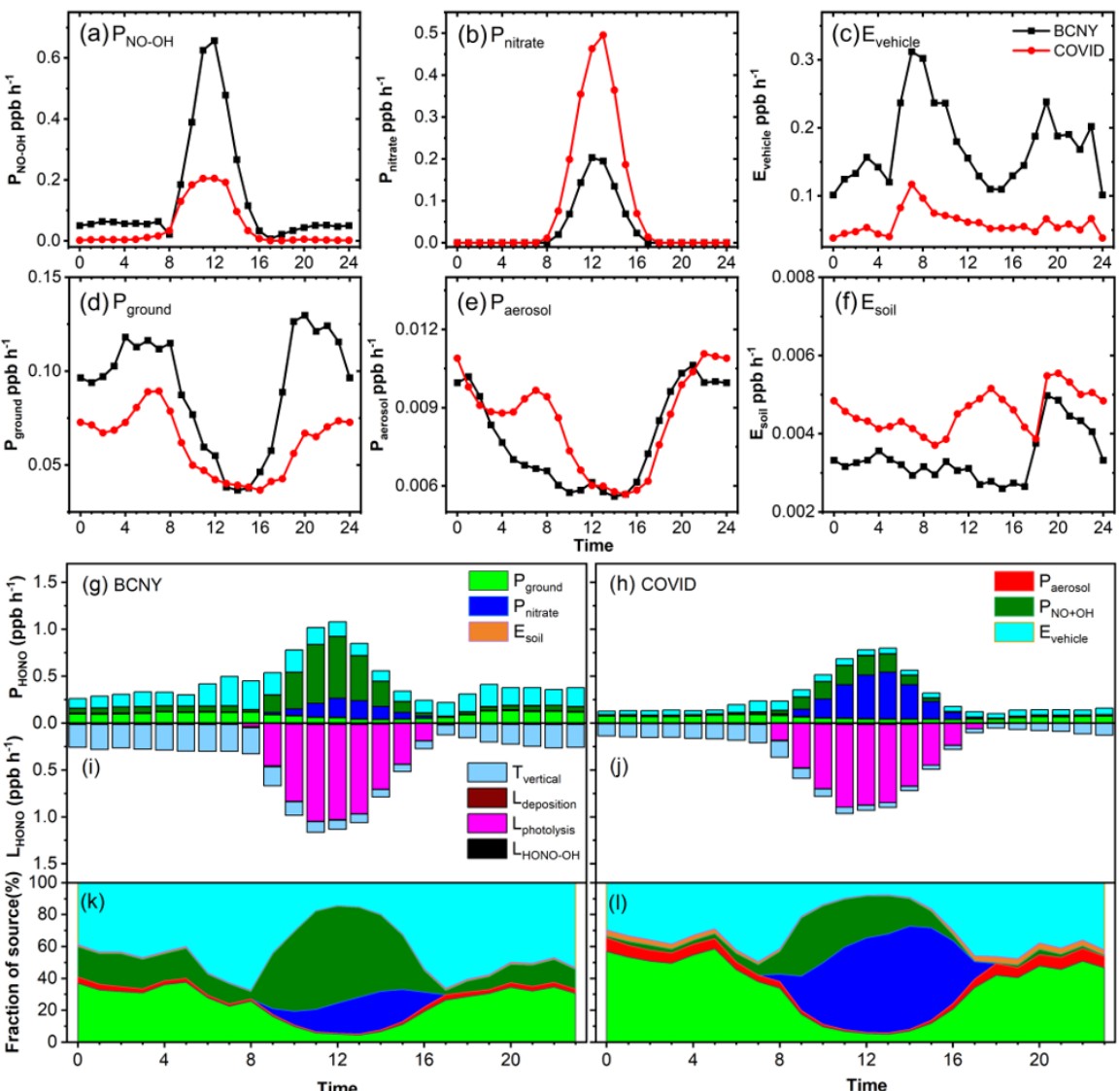


**Figure 4.** (**a-f**): Diurnal variations in HONO production rate from various sources. The black
lines are the diurnal curves before CNY and the red ones are during the COVID-19 lockdown.
(**g-l**): Variations of HONO budget. (**g,h**) Diurnal production rates of HONO; (**i,j**) loss rates of
HONO (unit: ppb h$^{-1}$); (**k,l**) relative contributions of each source. Panels (**g**),(**i**), and (**k**) show
the data from BCNY, and panels (**h**),(**j**), and (**l**) show the data from the COVID-19 lockdown.
The daytime HONO source related to photolysis of nitrate (0.223 ± 0.175 ppb h$^{-1}$) in the
P2 stage was much larger than that (0.107 ± 0.068 ppb h$^{-1}$) in the P1 stage. It contributed 16 ±
10% to the daytime HONO source in the P1 period. However, it became the most important
daytime source of HONO in the P2 stage, accounting for up to 53 ± 41%, as both the nitrate
concentration and the light intensity increased significantly ($P < 0.05$). Ye et al. (Ye et al., 2016)
reported that the photolysis rate constants of nitrate particles on the surface of different
materials were in the range of $6.0 \times 10^{-6}$ - $3.7 \times 10^{-4}$ s$^{-1}$. Thus, we used the lower limit value of
$6.0 \times 10^{-6}$ s$^{-1}$ and the upper limit value of $3.7 \times 10^{-4}$ s$^{-1}$ for sensitivity tests (methods M9 and M10),
which resulted in a change of 25% and 95% of HONO sources, respectively.
The direct emission rate of HONO from vehicles in the P1 stage was much higher than
that in the P2 stage. The emission rate of the P1 stage was between 0.135-0.39 ppb h$^{-1}$, with a
mean value of 0.227 ± 0.071 ppb h$^{-1}$. This is comparable with the value (0.079-0.32 ppb h$^{-1}$) in
the winter of 2018 (Liu et al., 2020d). In the P2 stage, it decreased to 0.062-0.173 ppb h$^{-1}$, with
a mean value of 0.086 ± 0.027 ppb h$^{-1}$. This value is slightly higher than the lower limit of
vehicle emissions of 0.013-0.076 ppb h$^{-1}$ estimated using an emission factor of 0.18% in our
previous study (Liu et al., 2020d), while it is less than the upper limit reported by Li (Li et al.,
2018) in Jinan of 0.13 ± 0.06-0.53 ± 0.23 ppb h$^{-1}$. During the lockdown, the emission rate of
HONO from vehicles was reduced by 53%-66% when compared with that before the lockdown.
In the P1 period, vehicle emission was an important nighttime source of HONO. It contributed
53 ± 17% to the HONO sources, much higher than heterogeneous reactions of NO$_2$ on aerosol
and ground surfaces (33%) (Fig. S9). In the P2 stage, due to the reduction of transport, the
contribution of vehicle emissions to HONO sources decreased to 40 ± 14%, while the
contribution of heterogeneous reactions of $NO_2$ increased to 53%. This is consistent with the
observed decrease in HONO concentrations. The daytime contributions of vehicle emissions
to HONO sources were lower than the corresponding nighttime contributions, while it was still
higher in the P1 period than in the P2 period. These results mean that vehicles should be
important contributors to ambient HONO under typical emission patterns in Beijing. In the
sensitivity analysis, the emission factors of 0.008 and 0.0186 were considered in methods M1
and M2, and 8% and 20% changes were found in the simulated HONO sources, respectively.
The yield of soil emissions in the P2 stage is also higher than that in the P1 stage due to the
temperature rise in the P2 stage because the temperature will affect the soil emission flux
(Oswald et al., 2013), while the importance of this source is negligible in this study. In M15
and M16, we amplify and shrink the soil emission flux by 10 times, respectively, and the change
of the simulated HONO sources was less than 5%.
As shown in Fig. 4e, the heterogeneous reaction rate of $NO_2$ on aerosols did not change
much between the P1 and P2 stages. The average production rate of HONO in the P1 stage was
$0.007 \pm 0.002$ ppb h$^{-1}$, and it was $0.008 \pm 0.002$ ppb h$^{-1}$ in the P2 stage, showing an increase of
about 14%. It is worth noting that the HONO formation rate from the heterogeneous conversion
of $NO_2$ on the surface of aerosol does not decrease, which is caused by the increase in $PM_{2.5}$
concentration along with a decrease in $NO_2$ concentration during the P2 period. If the
heterogeneous transformation of $NO_2$ on particulate surfaces is important, especially in the case
of heavy pollution, increased HONO concentrations should be expected instead of a large
decrease, as observed in the P2 stage. This is consistent with the changes in $HONO_{corr}/NO_2$ and
$PM_{2.5} \times NO_2$ as discussed in Sect. 3.2. For the heterogeneous transformation of $NO_2$ on the
ground and aerosol surfaces, this source is sensitive to the uptake coefficient ($\gamma$) of $NO_2$. For
the aerosol surface, here we assume that the upper limit of $\gamma$ is $10^{-5}$ (M7) and the lower limit is
$2\times10^{-7}$ (M8) (Liu et al., 2019b; Liu et al., 2020d). As shown in Table S3, the change in simulated
HONO is less than 5%. We reduced and magnified the surface area concentration ($A_s$) of
particulate matter by a factor of 10 in M11 and M12, respectively, and the resulting change in
HONO was still less than 10%. It should be noted that HONO is sensitive to the uptake
coefficient and surface area concentration. When the uptake coefficient is expanded by 5 times
or reduced by 10 times, the absolute HONO flux attributed to heterogeneous reactions increases
5 times or decreases 10 times, while the relative contribution is very low due to the small
absolute value of heterogeneous reactions compared with other sources.

Regarding the heterogeneous reaction of $NO_2$ on ground surfaces, the average formation

rate of HONO in the P1 stage was $0.09 \pm 0.03$ ppb h$^{-1}$, while it was $0.06 \pm 0.02$ ppb h$^{-1}$ in the
P2 stage. This is ascribed to the significant drop ($P < 0.05$) in $NO_2$ concentration during the
COVID-19 lockdown. Fig. 4k shows that the heterogeneous reaction of $NO_2$ on ground
surfaces is also an important nighttime source of HONO. In the P1 stage, heterogeneous
reactions on both aerosol and ground surfaces explained 33% of the nighttime HONO source.
In the daytime, however, the contribution of heterogeneous reactions to HONO sources
dropped rapidly. In the P2 stage, the heterogeneous reaction became the most important
nighttime source contributing up to 53% of HONO (Fig. S9). This can be explained by the
significant decrease ($P < 0.05$) in NO and direct emissions of HONO from traffic. Similar to
heterogeneous reactions on aerosol surfaces, we assumed that the upper limit of $\gamma_{NO2}$ on
ground surfaces was $10^{-5}$ (M5) and the lower limit was $2\times10^{-7}$ (M6), respectively, and the
changes in simulated HONO source were 40% and 9%, respectively. Indicating that HONO is
sensitive to the $NO_2$ uptake coefficient on the ground surface. In M13 and M14, we set the
surface roughness (δ) to 1 and 2.2 as reported in the literature, respectively (Zhang et al., 2022b;
Liu et al., 2020c), and the simulated changes in HONO were less than 8%.

During the P1 and P2 periods, the mean values of $T_{vertical}$ were $0.195 \pm 0.076$ ppb $h^{-1}$ and

$0.102 \pm 0.048$ ppb $h^{-1}$, respectively. It was the main sink of HONO at night. The mean $L_{photolysis}$
was $0.563 \pm 0.375$ ppb $h^{-1}$ and $0.442 \pm 0.324$ ppb $h^{-1}$, respectively, which was the main daytime
sink of HONO. The average loss rate of $L_{HONO\text{-}OH}$ during P1 and P2 was 0.005 ppb $h^{-1}$ and
0.004 ppb $h^{-1}$, respectively. The $L_{deposition}$ was $0.009 \pm 0.005$ ppb $h^{-1}$ during P1, while it was
$0.004 \pm 0.003$ ppb $h^{-1}$ during P2. In M17 and M18, we set the lower limit of the deposition rate
($V_d$) to 0.00077 and the upper limit to 0.025 (Zhang et al., 2023b), causing a change of 1% and
24% in the simulated HONO, respectively. In M19 and M20 at the same time, we set the
dilution rate ($K_{dilution}$) to 0.1 and 0.44, resulting in a 12% and 19% change, respectively.

It should be noted that each source is sensitive to the corresponding parameter as discussed

above. Thus, a more restrictive criterion is required to evaluate the reasonability of the
parameterization. We further estimated the HONO concentration according to Eq. (2) and the
parameters described in Sect. 2.2 to verify these calculated sources and sinks of HONO. Fig.
S6 shows the time series of estimated HONO concentrations. The observed HONO
concentrations were also shown for comparison. The estimated HONO concentrations were
well correlated with the observed values from the perspective of both diurnal curves and the
scattering point plot during the whole period (Fig. S7 and Fig. S8) although the estimated
HONO concentrations were slightly lower than the observed values at noon as shown in Fig.

S7. This means that our parametric scheme is overall reasonable but still underestimates the daytime HONO source due to some unknown daytime sources. This unknown source may be related to the photochemical reactions related to $NO_2$ and nitroaromatic compounds mentioned in recent studies (Liu et al., 2020a). Liu et al. have found the photoenhanced effect of the conversion from $NO_2$ to HONO on real urban grime and glass windows simulated in laboratory studies (Liu et al., 2019a; Liu et al., 2020b). Yang et al. also have proposed that photolysis of nitroaromatic compounds may be a daytime source of HONO (Yang et al., 2021). Considering the uncertainty of parameterization, we used Oracle Crystal Ball (version 11.1.2.4, Oracle's software for modeling, prediction, simulation, and optimization) (Rahmani et al., 2023) to evaluate the overall uncertainty of the parameterization through Monte Carlo simulations. The relative standard deviation is 27.2% for the HONO budget (details are in SI).

In summary, heterogeneous reactions of $NO_2$ (including ground and aerosol surfaces) contributed 33% to the nocturnal HONO sources in the P1 stage, while they increased to 53% in the P2 stage. Ground surfaces were the main interfaces for heterogeneous reactions, compared to aerosol surfaces. At the same time, vehicle emissions account for $53 \pm 17\%$ and $40 \pm 14\%$ of nighttime HONO sources in the P1 and P2 stages, respectively. To explore whether meteorological factors have an impact on the sources of HONO, we conducted the budget analysis of HONO using the deweathered pollutant concentrations. The results are shown in Fig. S10. When compared with the sources of HONO calculated using the raw concentration dataset (Fig. S9), it can be seen that deweathering has little effect on the daytime sources of HONO. For the nighttime source of HONO, however, deweathering caused the proportion of traffic emissions during BCNY increasing from 53% to 63% before the CNY or from 40% to

45% during the COVID-19 lockdown. The contribution of heterogeneous reactions of $NO_2$ on
ground surfaces decreased from 31% to 19% before the CNY or from 47% to 42% during the
COVID-19 lockdown. These results further highlight the importance of vehicle emissions to
nocturnal HONO sources in Beijing.

Therefore, regardless of whether the impact of meteorological conditions on the source of

HONO is considered, we can conclude that traffic-related emissions, rather than heterogeneous
reactions of $NO_2$ were the main HONO source at night in Beijing in the typical emission
patterns of air pollutants.
**4. Conclusions and atmospheric implications**.
During the COVID-19 pandemic at the beginning of 2020, the concentration of many air
pollutants decreased significantly ($P < 0.05$) due to the emission reduction of factories and
transportation. The average concentration of $NO_x$ decreased by about 57%, of which NO
decreased by about 87%, and $NO_2$ decreased by about 36%. The average concentration of
HONO decreased by about 45.3% compared with those before the pandemic control. The
average concentration of $O_3$ and $PM_{2.5}$ increased by approximately 75% and 50%, respectively.
It is worth noting that in addition to primary emissions, meteorological changes will also affect
changes in atmospheric pollutant concentrations. After removing meteorological factors, the
change proportions of $PM_{2.5}$ concentration in the two stages were -4.3% and -2.3% respectively.
The HONO changes were -8.3% and -3.8% respectively, the CO changes were -9.3% and -6.2%
respectively, and the $SO_2$ changes were +8.6% and +0.7% respectively. The change proportions
are all less than 10%, which means that the impact of changes in meteorological factors on
$PM_{2.5}$, HONO, CO, and $SO_2$ is very weak. However, the change proportions of NO in the two
stages were -16.2% and +32.8%, respectively, and $O_3$ was +39.6% and +6.2% respectively.
The change ratio is greater than 30%, indicating that NO and $O_3$ are greatly affected by
meteorology. In addition, the changes in $NO_2$ were -13.8% and -4.8% respectively, implying
that $NO_2$ is also affected by meteorological factors. From the entire observation period, except
for $O_3$, the changes of other species in the two periods fluctuated between 2.3% and 7.8% after
deweather, all less than 8%. In general, after removing the meteorological effects, NO
increased by 79%, $NO_2$ increased by approximately 29%, HONO decreased by approximately
43%, and $PM_{2.5}$ increased by approximately 50%. It is worth noting that $O_3$ increased by about
33%, which is much lower than the change in observed values (75.1%) (as shown in Table S6).
Although we have tried to assess the impact of meteorological factors quantitatively, this still
carries some uncertainty. In particular, uncertainty is inevitable for the source assessment of
substances such as HONO that are affected by a large number of parameters.
In this study, the parameters of HONO sources were optimized. The balance of sources
and sinks is well supported by a relatively high correlation between observed and estimated
HONO concentrations. During the observation period, we used lockdown during COVID-19
as a disturbance factor and compared the concentration and source changes of HONO before
and during COVID-19 lockdown to determine whether heterogeneous reactions on the surface
of particulate matter and vehicle emissions were important HONO sources. We found that
vehicle-related emissions were the most important nighttime HONO source in Beijing,
contributing 50-60% to the nighttime HONO sources. The homogeneous reaction between NO
and OH and the heterogeneous reaction of $NO_2$ on the aerosol surface were not important for
the contribution of nocturnal HONO, accounting for $13 \pm 5\%$ and $2 \pm 1\%$, respectively. The
heterogeneous reaction of $NO_2$ on ground surfaces was also found to be an important source of
HONO at night, accounting for $31 \pm 5\%$ of the nighttime HONO sources. Nitrate photolysis
became the most important source of HONO during the daytime compared with the situation
before the pandemic control because of the combined effect of the increase in the average
concentration of nitrate and the decrease in the NO concentration during the pandemic. We
conducted a potential source contribution function (PSCF, Fig S2) analysis in different periods,
i.e., BCNY and COVID, at the BUCT station and further compared the PSCF of HONO at
BUCT station with that at the Institute Atmospheric Physics (IAP) station, which is around 8
km from BUCT station, from January 24, 2022, to January 31, 2022, when the data were
available. The PSCF patterns were highly similar in different periods or locations. These results
mean that the air mass should be consistent during the COVID-19 lockdown and BCNY and
HONO should be evenly distributed in Beijing. Thus, the impact of meteorological changes on
the accuracy of observations cannot be ruled out, which is also a limitation of this study, but its
influence should be comparable between BCNY and the COVID lockdown. And the
conclusions drawn based on the observations at BUCT should represent the situation in Beijing.
Through uncertainty assessment, it was found that the assumption of $J_{NO3^-}$ would have the
greatest uncertainty, with a standard deviation of $\pm 19\%$. Nevertheless, this study confirms that
reducing anthropogenic emissions can indeed reduce the concentration of HONO in the
atmosphere. However, such reduction does not have a simple linear relation with the reduction
in human activities, but it also depends on meteorological conditions and complex chemical
transformation processes taking place in the atmosphere.

As a megacity in China, Beijing has a large population and intensive traffic emissions, as

a result of frequent air pollution. Although concentrations of HONO are usually lower than
those of other major pollutants, HONO efficiently triggers the formation of secondary
pollutants acting as an important primary source of OH radicals. Therefore, the sources of
HONO deserve to be investigated for air pollution control in Beijing. Our results suggest that
motor vehicle emissions are an important HONO source, while the contribution of the
heterogeneous conversion of $NO_2$ to HONO on the aerosol surfaces still needs to be further
evaluated and, especially, the kinetic parameters on ambient aerosol should be determined. In
future research, it is necessary to combine field observations, laboratory studies, and model
simulations to quantify the contribution of traffic-related emissions to HONO, and finally
obtain an accurate budget of HONO.
**Author contributions:** YZ contributed to the methodology, data curation, and writing of the
original draft. ZF and FZ contributed to the methodology, investigation, and data curation. CL
contributed to methodology, investigation, and data curation. WW contributed to the
conceptualization, investigation, reviewing, and editing the text, and supervision. XF
contributed to the methodology, reviewing, and editing the text. YZ and WM contributed to the
methodology, investigation, data curation, and reviewing and editing the text. ZL and CL
contributed to methodology, investigation, and data curation. GZ contributed to the
methodology, investigation, resources, and data curation. CY contributed to the methodology,
data curation, reviewing, and editing the text. VK, FB, TP, and JK contributed to the acquisition
of resources and reviewing and editing the text. MK contributed to the methodology and
reviewed the text. YL contributed to the conceptualization, investigation, data curation, writing,
reviewing & editing, supervision, and funding acquisition;
**Competing interests:** At least one of the (co-)authors is a member of the editorial board of
Atmospheric Chemistry and Physics.
**Data availability:** Data are available upon request to Yongchun Liu (liuyc@buct.edu.cn).
**Acknowledgments:** This research was financially supported by the Beijing Natural Science
Foundation (8232041), Beijing National Laboratory for Molecular Sciences (BNLMS-CXXM-
202011), and the National Natural Science Foundation of China (42275117 and 41931287).
This research was supported in part by Hebei Technological Innovation Center for Volatile
Organic Compounds Detection and Treatment in Chemical Industry (ZXJJ20220406) and the
Natural Science Foundation of Hebei Province (D2023209012). The work is partially
supported by Academy of Finland Flagship "Atmosphere and Climate Competence Center
(ACCC), project number 337549 and European Union's Horizon 2020 research and innovation
programme under grant agreement No 101036245 (RI-URBANS) and 101056783 (FOCI) as
well as Technology Industries of Finland Centennial Foundation via project "urbaani
ilmanlaatu 2.0". The authors would like to thank Dr. Zirui Liu from the Institute of Atmospheric
Physics, Chinese Academy of Sciences, for providing HONO dataset for comparison.

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
