# Peer review of "Concentration and source changes of HONO during the COVID-19 lockdown in Beijing"

_EGUsphere, 2023_

## Author Comment (AC1)

Dear Reviewer,

We appreciate your careful consideration of our manuscript. We have carefully responded to all of your **point-by-point** comments and issues and have revised the manuscript accordingly. These revisions are described in detail below.

**Reviewer #1**

The authors have made commendable efforts to conduct a comprehensive field measurement in urban Beijing, spanning the unique transition period from the pre-pandemic COVID-19 era to the subsequent lockdown period. While the controversies outlined in the introduction section (lines 95-122) remain unresolved, this study observes a notable decrease in HONO concentration from the pre-pandemic eve to the lockdown phase. Furthermore, the study offers a reasonable explanation regarding the contributions of HONO sources. I find this work particularly valuable as it underscores the significance of vehicle emissions and highlights the rapid temporal shifts in the roles of HONO sources, which could be of great interest to readers of ACP. I recommend publication of the manuscript following minor revisions. Below are my specific comments.

   **Response:** Thank you for your positive comments and good suggestions. We will respond to your comments point-by-point below.

1. The abstract requires improvement. Please remove lines 36-38 "which resulted in … to measure the HONO and related pollutants", and restructure the sentences for better clarity.

**Response:** Thank you for your suggestion. We have removed lines 36-38 "which resulted in the largest changes in air pollutant emissions in the history of modern atmospheric chemistry. A home-made Water-based Long-Path Absorption Photometer (LOPAP) along with other instruments were used to measure the HONO and related pollutants". In the revised manuscript, we reorganized them into "which resulted in a significant reduction in air pollutant emissions, providing a rare opportunity to understand the HONO budget in the atmosphere. We measured HONO and related pollutants" in lines 36-38.

2. Lines 67 and 80: The second and third paragraphs contain overlapping information. I recommend integrating the introduction about HONO concentrations into the first

paragraph. Subsequently, the second paragraph should focus solely on direct emissions, while the third paragraph should discuss secondary formation.

**Response:** Thank you for your good suggestion. We have integrated the introduction of HONO concentrations into the first paragraph, the original "Intensive studies have been carried out on HONO measurements and source analysis (Liu et al., 2020c; Liu et al., 2020d; Zheng et al., 2020; Zhang et al., 2020; Xue et al., 2020; Zhang et al., 2019a; Liu et al., 2019b). The concentrations of HONO in the atmosphere range from a few ppt in remote areas (Spataro et al., 2016) to several ppb, even several tens ppb in heavily polluted areas (Liu et al., 2019b; Liu et al., 2020c; Liu et al., 2020d; Zheng et al., 2020)." in lines 63-67 has been moved to lines 61-66 in the revised manuscript.

As you suggested, the second paragraph only focuses on direct emissions in lines 68-79 in the revised manuscript "Direct emissions include soils, biomass burning, vehicles, indoor air, and livestock farming. Soil emissions, which depend on soil types, microorganisms, water content, temperature, and pH (Kulmala and Petäjä, 2011; Weber et al., 2015; Kim and Or, 2019), are important sources of HONO. Biomass burning, often occurs in the summer and autumn when wheat/corn is harvested and wildfires are common (Zhang et al., 2019b; Sun et al., 2017; Sun et al., 2018; Peng et al., 2020). Vehicle emissions are considered an important source of HONO in traffic-intensive areas (Kramer et al., 2020; Li et al., 2021). This source is more important at nighttime compared with daytime (Zhang et al., 2016; Fu et al., 2019; Liu et al., 2020d). Recently, indoor emissions have also been proposed as a potential HONO source (Xue, 2022), which is related to the ventilation from high HONO concentrations in indoor air to low HONO concentrations in outdoor air (Zhang et al., 2019b). Livestock farming is a previously overlooked source of HONO, especially in agricultural areas."

The third paragraph discusses secondary formation in lines 80-88 in the revised manuscript "Secondary formation of HONO includes gas-phase reaction between NO and OH radicals, photolysis of particulate nitrate, and heterogeneous reaction of $NO_2$ on ground and particulate matter surfaces, including photochemical heterogeneous reaction of $NO_2$. Gas phase reaction between NO and OH, photolysis of nitrate particles, and light-enhanced conversion of $NO_2$ are the main daytime sources of HONO (Liu et al., 2019c; Liu et al., 2020d; Zhang et al., 2022b). Furthermore, acid replacement processes may be a non-negligible source of daytime HONO in locations affected by soil-borne mineral dust deposition (Vandenboer et al., 2014). The heterogeneous

reaction of $NO_2$ on various surfaces is widely regarded as an important source of HONO (Han et al., 2016; Liu et al., 2020b).".

3. Line 71: Here, "harvest season" may not be the most accurate term. Biomass burning encompasses a range of activities, including wildfires, which can be particularly significant under certain circumstances.

**Response:** Thank you for your good suggestion. We have modified it in the revised manuscript to "Biomass burning, often occurs in the summer and autumn when wheat/corn is harvested and wildfires are common (Zhang et al., 2019b; Sun et al., 2017; Sun et al., 2018; Peng et al., 2020)" in lines 71-73.

4. Line 85: The authors omitted the acid displacement (VandenBoer et al. 2015, Liu et al. 2019)?

**Response:** Thank you for your good suggestion. In lines 85-86 in the revised manuscript, we added a sentence "Furthermore, acid replacement processes may be a non-negligible source of daytime HONO in locations affected by soil-borne mineral dust deposition (Vandenboer et al., 2014)".

5. Line 79: Livestock HONO emission is negligible in urban Beijing. Similarly, the emission of HONO from soil, which may peak following fertilizer application, likely does not need to be considered.

**Response:** Thank you for your good suggestion. The HONO emissions from Livestock in urban Beijing can be ignored, while soil emissions are slightly more important than Livestock. In this study, the contribution of the soil is even comparable to that of heterogeneous conversion of $NO_2$ on the surfaces of aerosol. Therefore, we still include the soil emissions when performing the budget analysis.

6. Line 263: heterogeneous yield of HONO instead of $NO_2$?

**Response:** Thank you for your good suggestion. In the revised manuscript in line 263, we changed it to "heterogeneous yield of HONO".

7. Section 3.1 Please present the result following the order of Figure 1 (a) ~ (i).

**Response:** Thank you for your good suggestion. In Figure 1, we moved the

meteorological parameters to the end of this figure. And revised the description order of pollutants accordingly in lines 318-341 in the revised manuscript.

8. Line 406: How about the heterogenous conversion on ground?

**Response:** Thank you for your good comments. The description here is inaccurate, in the revised manuscript, we corrected the sentence "These results imply that HONO might be more influenced by vehicle emissions than by heterogeneous reactions on aerosol surfaces" in lines 405-407.

9. Line 236: Is $HONO_{corr}$ equivalent to $C_{HONO,corr,t}$ as indicated in Eq. (6)? Please maintain consistency in the use of abbreviations.

**Response:** Thank you for your good comments. In Section 2.2, to maintain the consistency of the formula, we use "$C_{HONO,corr,t}$", and in subsequent chapters, for the convenience of readers, we use "$HONO_{corr}$". In order not to mislead readers, in the revised manuscript, we use $HONO_{corr}$ to represent it uniformly.

10. Line 508: In fact, Song et al. (2023) proposed two HONO production pathways distinct from the homogeneous reaction between NO and OH.

**Response:** Thank you for your good suggestion. The quote here is inappropriate, in lines 508-512 in the revised manuscript, we have modified it to "This is consistent with previous studies in urban Beijing (Gu et al., 2021; Jia et al., 2020; Liu et al., 2021). Interestingly, a recent study proposed a new mechanism through smog chamber experiments, that is, NOx photooxidation (reaction of NO and adsorbed $HNO_3$) may be an important daytime HONO source (Song et al., 2023), although it has not yet been verified by field observations."

11. Line 534: "a change of 25% and 95% of HONO sources, respectively." What does this signify? It appears that the selection of the photolysis rate constant could have a significant influence.

**Response:** Thank you for your good comment. We agree with you that the selection of the photolysis rate constant has a significant influence on its contribution to the HONO source. The big difference in the estimated source here means both the upper and lower limits of the parameters might be improper. Thus, it requires a more restrictive method to evaluate the reasonability of the parameterization and assess the

overall uncertainty. In our study, we further compared the estimated hourly HONO concentrations with the observed values to evaluate the parameters finally chosen. In lines 605-608 in the revised manuscript, we added a short paragraph "It should be noted that each source is sensitive to the corresponding parameter as discussed above. Thus, a more restrictive criterion is required to evaluate the reasonability of the parameterization. We further estimated the HONO concentration according to Eq. (2) and the parameters described in Sect. 2.2 to verify these calculated sources and sinks of HONO". In addition, the overall uncertainty is 27.2% for the HONO budget evaluated based on Monte Carlo simulations. This has been pointed out in line 624 in the revised manuscript.

12. Lines 532-537: Move the results regarding soil emissions to the subsequent paragraph.

**Response:** Thank you for your good suggestion. We have moved the paragraph "The yield of soil emissions in the P2 stage is also higher than that in the P1 stage due to the temperature rise in the P2 stage because the temperature will affect the soil emission flux (Oswald et al., 2013), while the importance of this source is negligible in this study. In M15 and M16, we amplify and shrink the soil emission flux by 10 times, respectively, and the change of the simulated HONO sources was less than 5%." to the subsequent paragraph in lines 554-559 in the revised manuscript.

13. Line 564: The $NO_2$ concentration decreased significantly from 26.9 ppb to 17.2 ppb (-36%) from P1 to P2, indicating a substantial reduction rather than a slight decrease.

**Response:** Thank you for your good suggestion. In lines 563-565 in the revised manuscript, we revised it as "It is worth noting that the HONO formation rate from the heterogeneous conversion of $NO_2$ on the surface of aerosol does not decrease, which is caused by the increase in $PM_{2.5}$ concentration along with a decrease in $NO_2$ concentration during the P2 period".

14. Lines 570-573: It's noteworthy that HONO exhibits minimal sensitivity to both the uptake coefficient (γ) and surface area concentration (As). However, the authors should provide an explanation for this phenomenon.

**Response:** Thank you for your good comments. According to eq(7), the contribution of the heterogeneous reaction of $NO_2$ on aerosol surfaces to the HONO

source should be sensitive to both the uptake coefficient ($\gamma$) and surface area concentration ($A_s$). Although the absolute values change prominently as expected when the $A_s$ or $\gamma$ is increased or reduced, its relative contribution does not change obviously due to the small absolute value compared with other sources.

In lines 575-579 in the revised manuscript, we have revised the paragraph "It should be noted that HONO is sensitive to the uptake coefficient and surface area concentration. When the uptake coefficient is expanded by 5 times or reduced by 10 times, the absolute HONO flux attributed to heterogeneous reactions increases 5 times or decreases 10 times, while the relative contribution is very low due to the small absolute value of heterogeneous reactions compared with other sources."

15. Line 593: These variations (-9% to +40%) are significant and warrant attention.

**Response:** Thank you for your good comments. As mentioned in the previous reply, this sensitivity is closely related to the contribution of heterogeneous transformation of $NO_2$ at the ground surface to HONO. When the uptake coefficient is expanded by 5 times or reduced to 0.1, HONO changes by 40% and 9% respectively, indicating that HONO is sensitive to the $NO_2$ uptake coefficient on the ground surface, implying the importance of this source. In the revised manuscript, we added the sentence "Indicating that HONO is sensitive to the $NO_2$ uptake coefficient on the ground surface." in lines 592-593.

16. Lines 622: The method utilized to estimate the overall uncertainty of the parameterization should be presented in the Experimental section.

**Response:** Thank you for your good suggestion. In lines 296-298 in the revised manuscript, we added a sentence "Oracle Crystal Ball (version 11.1.2.4, Oracle's software for modeling, prediction, simulation, and optimization) (Rahmani et al., 2023) to evaluate the overall uncertainty of the parameterization through Monte Carlo simulations. The details are shown in Text S2 in the SI". In the revised SI, we added an introduction to the Monte Carlo algorithm as follows:

**Text S2 Monte Carlo algorithm**

The Monte Carlo algorithm is a method of estimating numerical values through random sampling. It can be used to estimate the overall uncertainty of the numerical value. A large number of samples are generated by random sampling from a probability distribution and the required numerical indicators are calculated based on these samples.

Due to the limited number of samples, there is a certain error between the estimated value and the true value. We increase the number of sampling times to 10,000 to reduce statistical uncertainty.

When establishing the simulation model, the respective change ranges of the variables that affect HONO intensity are input, and the uncertainty of the modeling is evaluated by sampling from the probability distribution of the parameters to obtain the overall uncertainty. In addition, the uncertainty of the model parameters is propagated to the model output through Monte Carlo sampling, and the uncertainty distribution of the results can be obtained. The formula for overall uncertainty can be expressed as:

$$\sigma = \sqrt{\frac{1}{N}\sum_{i=1}^{N}(x_i - \bar{x})^2}$$

σ represents the standard deviation of the overall uncertainty; N is the number of samples; $x_i$ is the value of the $i^{th}$ sample, and $\bar{x}$ is the mean of the sample.

References:

Gu, R., Shen, H., Xue, L., Wang, T., Gao, J., Li, H., Liang, Y., Xia, M., Yu, C., Liu, Y., and Wang, W.: Investigating the sources of atmospheric nitrous acid (HONO) in the megacity of Beijing, China, Science of The Total Environment, 10.1016/j.scitotenv.2021.152270, 2021.

Liu, Y., Nie, W., Xu, Z., Wang, T., Wang, R., Li, Y., Wang, L., Chi, X., and Ding, A.: Semi-quantitative understanding of source contribution to nitrous acid (HONO) based on 1 year of continuous observation at the SORPES station in eastern China, Atmospheric Chemistry and Physics, 19, 13289-13308, 10.5194/acp-19-13289-2019, 2019a.

Liu, Y., Ni, S., Jiang, T., Xing, S., Zhang, Y., Bao, X., Feng, Z., Fan, X., Zhang, L., and Feng, H.: Influence of Chinese New Year overlapping COVID-19 lockdown on HONO sources in Shijiazhuang, Sci Total Environ, 745, 141025, 10.1016/j.scitotenv.2020.141025, 2020a.

Liu, Y., Lu, K., Li, X., Dong, H., Tan, Z., Wang, H., Zou, Q., Wu, Y., Zeng, L., Hu, M., Min, K. E., Kecorius, S., Wiedensohler, A., and Zhang, Y.: A Comprehensive Model Test of the HONO Sources Constrained to Field Measurements at Rural North China Plain, Environ Sci Technol, 53, 3517-3525, 10.1021/acs.est.8b06367, 2019b.

Liu, Y., Zhang, Y., Lian, C., Yan, C., Feng, Z., Zheng, F., Fan, X., Chen, Y., Wang, W., Chu, B., Wang, Y., Cai, J., Du, W., Daellenbach, K. R., Kangasluoma, J., Bianchi, F., Kujansuu, J., Petäjä, T., Wang, X., Hu, B., Wang, Y., Ge, M., He, H., and Kulmala, M.: The promotion effect of nitrous acid on aerosol formation in wintertime in Beijing: the possible contribution of traffic-related emissions, Atmospheric Chemistry and Physics, 20, 13023-13040, 10.5194/acp-20-13023-2020, 2020b.

Shi, X., Ge, Y., Zheng, J., Ma, Y., Ren, X., and Zhang, Y.: Budget of nitrous acid and its impacts on atmospheric oxidative capacity at an urban site in the central Yangtze River Delta region of China,

Atmospheric Environment, 238, 10.1016/j.atmosenv.2020.117725, 2020.

Song, Y., Zhang, Y., Xue, C., Liu, P., He, X., Li, X., and Mu, Y.: The seasonal variations and potential sources of nitrous acid (HONO) in the rural North China Plain, Environ Pollut, 311, 119967, 10.1016/j.envpol.2022.119967, 2022.

Zhang, J., An, J., Qu, Y., Liu, X., and Chen, Y.: Impacts of potential HONO sources on the concentrations of oxidants and secondary organic aerosols in the Beijing-Tianjin-Hebei region of China, Sci Total Environ, 647, 836-852, 10.1016/j.scitotenv.2018.08.030, 2019a.

Zhang, J., Chen, J., Xue, C., Chen, H., Zhang, Q., Liu, X., Mu, Y., Guo, Y., Wang, D., Chen, Y., Li, J., Qu, Y., and An, J.: Impacts of six potential HONO sources on HOx budgets and SOA formation during a wintertime heavy haze period in the North China Plain, Sci Total Environ, 681, 110-123, 10.1016/j.scitotenv.2019.05.100, 2019b.

Breiman, L.: Random Forests, Machine Learning, 45, 5-32, https://doi.org/10.1023/A:1010933404324, 2001.

Grange, S. K. and Carslaw, D. C.: Using meteorological normalisation to detect interventions in air quality time series, Sci Total Environ, 653, 578-588, 10.1016/j.scitotenv.2018.10.344, 2019.

Grange, S. K., Carslaw, D. C., Lewis, A. C., Boleti, E., and Hueglin, C.: Random forest meteorological normalisation models for Swiss $PM_{10}$ trend analysis, Atmospheric Chemistry and Physics, 18, 6223-6239, 10.5194/acp-18-6223-2018, 2018.

---

## Author Comment (AC2)

Dear Reviewer,

We appreciate your careful consideration of our manuscript. We have carefully responded to all of your **point-by-point** comments and issues and have revised the manuscript accordingly. These revisions are described in detail below.

**Reviewer #2**

This paper presents an analysis of measurements of gas-phase nitrous acid (HONO), and related air pollutants/atmospheric chemical species, performed in the boundary layer in Beijing prior to and during the Covid-19 lockdowns in early 2020. The variation in abundance of HONO, its precursors and related species are used to quantify different chemical source and sink terms for HONO, and from their diurnal variation and change across the start of lockdown, inferences drawn re their relative importance – in particular direct vehicle emission.

The abundance and sources of HONO are very much a current topic in urban atmospheric chemistry/air pollution, as besides being a pollutant in its own right HONO is usually the major precursor to (reservoir for) the key oxidant OH. Lockdown presents a unique near-step-change opportunity to explore these processes. This paper adds to the developing understanding of these sources, and in that respect is a valuable contribution to the literature.

The measurements appear to have been carefully performed and are clearly presented, and the analysis is interesting, with clear signal in the change in mean diurnal variations (which are used to develop qualitative arguments).

However, I have substantial reservations about several key assumptions/aspects of the quantitative analysis approach, which in my opinion do not allow the authors to draw the conclusions they reach. These are:

    **Response:** Thanks for your positive comments and good suggestions. We will reply to your concerns point-by-point below.

**Major comments**

**1. Meteorology.** Obviously, weather changes affect pollutant advection, import,

dispersion, chemical processing, etc. The paper presents "before" vs "after" comparisons of concentrations either side of the start of lockdown, without consideration (beyond a basic vertical dispersion parameterization) of these effects. There is significant literature on the importance of correcting for meteorology–for example applying de-weathering approaches – several using the specific example of air pollutant abundance in Beijing across lockdown (e.g. Jiabo et al., 2021; Shi et al., 2021; Lv et al., 2022). How much of the observed change (or lack of change) in each species is due to changes in the weather between the two time periods P1 and P2? It would be interesting to repeat the analysis using deweathered concentrations.

**Response:** Thank you for your good comments and suggestions. We agree with you that meteorology is always one of the important factors affecting the concentration of air pollutants. As you suggested, we performed the de-weather analysis using a Random Forest model with a deweather module.

In lines 345-369 in the revised manuscript, we added a paragraph "It is worth noting that changes in atmospheric pollutant concentrations are affected by both emissions and meteorology. Especially, during the lockdown period, meteorological conditions in Beijing were not conducive to the dispersion of pollutants, thus the impact of meteorological conditions on the concentration of these pollutants needs to be assessed. We use the random forest algorithm of machine learning to remove the influence of meteorology from air quality time series data by a deweather method. The details are present in Text S1 in the SI. The model performs well in predicting the concentrations of pollutants compared to the observations in both the training and test datasets (Table S5). The concentrations and relative changes of each pollutant after deweather are recorded in Table S6. After deweather, the mean concentration of $PM_{2.5}$ increased significantly from 45.22±28.56 in P1 to 67.92±57.97 μg m$^{-3}$ in P2 at a confidence level of 0.05, with an increase of 50.2%; The mean concentration of HONO was 0.89±0.37 ppb in P1, while it decreased to 0.51±0.25 ppb in P2, with a drop of 42.70%; The concentrations of NO and $NO_2$ significantly decreased from 15.44±18.40 and 23.28±7.28 ppb in P1 to 3.24±2.05 and 16.43±5.98 ppb in P2, respectively, which decreased by 79.02% and 29.42% respectively; $SO_2$ decreased from 2.27±0.69 in P1 to

1.48±1.18 ppb in P2, a decrease of approximately 34.8%; CO increased from 823.60±318.92 in P1 to 896±488.29 ppb in P2 (an increase of 8.79%) and $O_3$ increased from 16.98±5.62 to 22.60±4.10 ppb, an increase of about 33.1%, which was much lower than the change range of observed values (75.08%). As shown in Table S6, meteorological conditions have a significant impact on $O_3$ concentration. The impact was +39.64% and +6.15% in P1 and P2, respectively. The impact of deweather on NO in the two periods was -16.18% and +32.79%, respectively. It was -13.75% and -4.81%, respectively, for $NO_2$. However, the changes of other species in the two periods after deweather fluctuated between 2.3% and 7.8%. This implies that meteorological conditions have an important impact on the concentrations of NO and $O_3$, while meteorological factors have little impact on HONO, $SO_2$, CO and $PM_{2.5}$."

In lines 421-423 in the revised manuscript, we added the sentence "After deweather, the HONO concentration decreased significantly from 0.89±0.37 in P1 to 0.51±0.25 ppb in P2 at a confidence level of 0.05, with a decrease of 42.7%. This means that meteorology has little impact on HONO."

In the revised SI, we added the details about the random forest model to correct the influence of meteorology, including an introduction to the algorithm, a flow chart, and a model evaluation. The added content is as follows:

**Text S1 De-weather model**

Changes in atmospheric pollutant concentration are affected by emissions and meteorology. Machine learning models, including boosted regression trees and random forest (RF) algorithms, often exhibit higher predictive accuracy because of their advantages in modeling complex relationships between response variables and predictor variables (Zhan et al., 2018). By reducing the variance/bias and error of high-dimensional data sets, it has better performance compared to traditional statistical and air quality models. The algorithm resolves the relationship between air pollutant levels and their predictors, including meteorological parameters and time variables such as the day of the year (Julian Day), day of the week (Monday to Sunday), and hour of the day (0-23) (Grange et al., 2018). The input data set was randomly divided into a training data set for building the RF model (i.e., 70% of the input data set) and a testing data set

(30% of the input data set) for testing the performance of the RF model using unseen data sets. The RF model is an ensemble model composed of many individual decision tree models (Breiman, 2001).

In the RF model, the bagging algorithm is utilized, which involves randomly selecting samples from the training dataset, with replacement, along with their respective predictor features. Each decision tree is grown based on various decision rules that optimize the fitting between observed pollutant concentrations (response variable) and their predictor features. The selection of predictor features for each tree node is performed randomly to achieve the best possible split. The predicted pollutant concentrations are determined by aggregating the outcomes of all individual decision trees through a weighted average. The bagging process, by averaging predictions from bootstrap samples, helps reduce variance and mitigates overfitting issues in the model. As shown in Figure S1, the entire data set is randomly divided into two groups, one is the training data set, used to build the random forest model; the other is the test data set, used for testing without seeing the data set. The training data set accounts for 70% of the total data, and the rest is test data. Grange et al. (2018) built the RF model using the R "normalweather" package.

In our study, the parameters of the RF model are as follows: hourly concentrations of HONO, NO, $NO_2$, $O_3$, $PM_{2.5}$, $SO_2$, and CO as dependent variables, meteorological parameters (wind direction, wind speed, air temperature, humidity, and atmospheric pressure) and Time predictors (weekdays, hours) served as independent predictors. The training set uses randomly selected 70% of the data, and the remaining 30% is used as the test set. Random forest models were developed using the rmweather R package (Grange et al., 2018; Grange and Carslaw, 2019). The number of trees is 300, and the number of variables split in each node is 3. For each weather normalization, the explanatory variables are resampled (excluding the time variable) without replacement and randomly assigned to the dependent variable observations. The 1000 predicted values are then aggregated using the arithmetic mean to obtain the deweathered concentration.

[Figure]

**Fig. S1.** The flowchart of the machine learning based RF algorithm.

**Model performance evaluation**

Evaluation metrics for the model can be found in Table S5. The random forest model showed good performance in predicting the data compared to the observations in the training and test datasets. Specifically, the R values range from 0.93-0.98. These extremely high correlation values indicate a strong relationship between the predicted values and the observed values, indicating that the characteristics of the established model are excellent. The FAC2 of each indicator is very small, indicating that our model meets the conditions for predicting scores. Likewise, lower NMB and NMGE values indicate that our model performs well. Through the verification of various indicators, it is believed that the model has good prediction ability.

**Table S5**. RF model performance for testing data set (in hourly time resolution).

| Pollutants | RMSE | R | FAC2 | MB | MGE | NMB | NMGE |
|---|---|---|---|---|---|---|---|
| HONO | 0.21 | 0.93 | 0.86 | 0.01 | 0.15 | 0.02 | 0.21 |
| NO | 7.30 | 0.93 | 0.34 | -0.21 | 3.76 | -0.03 | 0.50 |
| $NO_2$ | 4.38 | 0.94 | 0.93 | -0.04 | 3.12 | 0.00 | 0.16 |

| | | | | | | | |
|---|---|---|---|---|---|---|---|
| O$_3$ | 4.04 | 0.95 | 0.84 | 0.12 | 2.91 | 0.01 | 0.16 |
| SO$_2$ | 0.63 | 0.93 | 0.68 | 0.01 | 0.38 | 0.01 | 0.27 |
| CO | 164.55 | 0.96 | 1.00 | 4.22 | 114.60 | 0.00 | 0.13 |
| PM$_{2.5}$ | 12.88 | 0.98 | 0.88 | 0.83 | 8.70 | 0.01 | 0.15 |

Note: FAC2 (fraction of predictions with a factor of two), MB (mean bias), MGE (mean gross error), NMB (normalized mean bias), NMGE (normalized mean gross error), COE (Coefficient of Efficiency), IOA (Index of Agreement).

In the revised SI, we have added Table S6(Table R2). After deweather, the concentration changes of different pollutants in the two time periods will be introduced in detail in the subsequent replies.

**Table R2.** Periods and concentration after deweather (mean ± standard deviation) of PM$_{2.5}$, HONO, trace gases in field observation, and the percentages in parentheses are concentration changes after deweather. Relative change in observed values and deweather values in different periods.

| Category | BCNY (1.1-1.24) | | COVID (1.25-3.6) | | Relative change | |
|---|---|---|---|---|---|---|
| | Deweather | Observed | Deweather | Observed | Deweather | Observed |
| PM$_{2.5}$ (μg/m$^3$) | 45.22±28.56 (-4.26%) | 47.23±44.50 | 67.92±57.97 (-2.28%) | 69.86±67.26 | +50.20% | +47.91% |
| HONO (ppb) | 0.89±0.37 (-8.25%) | 0.97±0.74 | 0.51±0.25 (-3.77%) | 0.53±0.45 | -42.70% | -45.36% |
| NO (ppb) | 15.44±18.40 (-16.18%) | 18.42±29.24 | 3.24±2.05 (+32.79%) | 2.44±5.40 | -79.02% | -86.75% |
| NO$_2$ (ppb) | 23.28±7.28 (-13.75%) | 26.99±13.41 | 16.43±5.98 (-4.81%) | 17.26±11.34 | -29.42% | -36.05% |
| CO (ppb) | 823.60±318.92 (-9.27%) | 907.72±499.16 | 896±488.29 (-6.17%) | 954.87±624.04 | +8.79% | +5.19% |
| SO$_2$ (ppb) | 2.27±0.69 (+8.61%) | 2.09±1.36 | 1.48±1.18 (+0.68%) | 1.47±1.95 | -34.80% | +29.67% |
| O$_3$ (ppb) | 16.98±5.62 (+39.64%) | 12.16±10.79 | 22.60±4.10 (+6.15%) | 21.29±11.78 | +33.10% | +75.08% |

**2. Chemical lifetimes.** The paper in effect performs a steady state analysis on HONO,

assuming any rate of change of concentration can be related to an imbalance in the in-situ source and sink terms. You can do this for short-lived species – such as OH – but only with care for species such as HONO, with lifetimes (these are midlatitude winter measurements) of tens of minutes. The measured HONO reflects the integrated chemical variability over the sampled airmass trajectory prior to its arrival at the measurement point. This will be heterogeneous – especially at ground level in the middle of a city (i.e. the OH and $NO_2$, etc will have varied a lot over the period of time – given by the HONO lifetime – that the airmass has traveled to the measurement point within the city). See arguments developed by Lee et al, JGR 2013, and related papers.

**Response:** Thank you for your good comments and suggestions. We agree with you that the measured HONO reflects the integrated chemical variability over the sampled airmass trajectory prior to its arrival at the measurement point. This might introduce uncertainty about the budget analysis using a steady state analysis. To confirm the possible affected areas, we carefully checked the representativeness of the dataset. Firstly, a potential source contribution function (PSCF) of HONO has been analyzed. Given a 10 min lifetime of HONO, the spatial distribution of HONO in the P2 is highly similar to that when compared with that in the P1 as shown in Figure R2. This means air mass should have a similar impact on the HONO budget during these two periods. Secondly, we carried out deweather analysis using a random forest algorithm combined with a machine learning model. The details are shown in the SI. We found meteorology has little influence on HONO concentration during the whole observation period. This means HONO concentration is dominated by primary emissions and secondary formation. Thirdly, we compared the concentrations of HONO at BUCT and Institute of Atmospheric Physics (IAP) from Jan. 5[th] to Mar 24[th] when the data are available (Figure R3). The linear distance between these two sites is ~8 km. The HONO concentrations are comparable at these two sites. Thus, even though the concentration changes represent the integrated chemical variability over the sampled airmass trajectory prior to its arrival at the measurement point, it still represents the local feature in urban Beijing. Finally, the instrumentation time resolution of LOPAP was 6 s. We calculated the variation coefficient for the datasets with different time resolution, i.e., 1

h *vs* 6 s. A small variation coefficient of ~0.02-0.05 implies that a small uncertainty of HONO budget might be resulted from the lifetime of HONO. Thus, we think the possible uncertainty should not have a large influence on our conclusions when the budget is compared at a fixed site between two different periods.

In lines 206-215 in the revised manuscript, we added a short paragraph to clarify this possible uncertainty as "Given that the result of potential source contribution function (PSCF, Fig S2), the source distribution of HONO between BCNY and COVID was highly similar and the trend of HONO was similar (Pearson'r=0.78) between BUCT and Institute of Atmospheric Physics (IAP, 8 km away from BUCT), the steady state analysis on HONO is appliable and reasonable even though the lifetime of HONO is several minutes in the atmosphere. In addition, the instrumentation time resolution of LOPAP was 6 s. We calculated the variation coefficient for the datasets with different time resolutions, i.e., 1 h *vs* 6 s. A small variation coefficient of ~0.02-0.05 implies that a small uncertainty of the HONO budget might result from the lifetime of HONO. Thus, we think the possible uncertainty should not have a large influence on our conclusions when the budget is compared at a fixed site between two different periods". And we have added Figure R2 as Figure S2 in the revised SI.

[Figure]

**Figure R2.** The potential source contribution function (PSCF) maps for the concentration of HONO (a and b are BCNY and COVID, respectively). The trajectory of the air mass is 12 hours.

[Figure]

**Figure R3.** The comparison of HONO concentration between IAP and BUCT from Jan. 5$^{th}$ to Mar 24$^{th}$.

**3. Statistical** significance/precision/uncertainty. The paper does not consider sufficiently the uncertainty in the (many) source and sink terms considered, and their propagation together (beyond the Monte Carlo result, which I can't believe includes the uncertainty in the individual inputs, e.g. OH concentrations). Is there any statistical power resulting from their combined uncertainties? Are the uncertainties given in the paper 1 or 2 standard deviations? Are differences statistically significant? The ± ranges – even assuming these are 1 sd – suggest not (e.g. HONO/NO$_2$ ratio – changing from 0.038 ± 0.035 to 0.042 ± 0.034 – this is not a meaningful (statistically significant) change). There are several examples.

   **Response:** Thank you for your comments and suggestions. As for your questions, we will reply one by one.

   In our calculation, ten key parameters including source terms and sink terms were considered, and their respective fluctuation ranges were input, and then Monte Carlo was used to calculate the respective sensitivity distributions and uncertainties. In

addition, through comparison with field observations in the Beijing area, we found that our OH fitting performance is good. Of course, it should be noted that when the fluctuation range of various parameters were input, this may still bring some deviations although we have given full consideration and optimized based on existing research. Considering that the sources and sinks of HONO are complex and the selection of parameters in different studies may even differ by several orders of magnitude, our results should generally perform well.

For the comprehensive uncertainty, the uncertainty given in the paper is 1 standard deviation. In the revised manuscript, we added a description where it first appears "During P1, the measured concentration of $PM_{2.5}$ varied between 0.2-288 $\mu g\ m^{-3}$ and the mean concentration was $47.2 \pm 44.5$ (mean $\pm\ 1\sigma$) $\mu g\ m^{-3}$." in lines 318-319. And we added the sentence "The relative standard deviation is 27.2% for the HONO budget (details are in SI)" in line 624. In the revised SI, we added an introduction to the Monte Carlo algorithm. The added content is as follows:

**Text S2 Monte Carlo algorithm**

The Monte Carlo algorithm is a method of estimating numerical values through random sampling. It can be used to estimate the overall uncertainty of the numerical value. A large number of samples are generated by random sampling from a probability distribution and the required numerical indicators are calculated based on these samples. Due to the limited number of samples, there is a certain error between the estimated value and the true value. We increase the number of sampling times to 10,000 to reduce statistical uncertainty.

When establishing the simulation model, the respective change ranges of the variables that affect HONO intensity are input, and the uncertainty of the modeling is evaluated by sampling from the probability distribution of the parameters to obtain the overall uncertainty. In addition, the uncertainty of the model parameters is propagated to the model output through Monte Carlo sampling, and the uncertainty distribution of the results can be obtained. The formula for overall uncertainty can be expressed as:

$$\sigma = \sqrt{\frac{1}{N}\sum_{i=1}^{N}(x_i - \bar{x})^2}$$

σ represents the standard deviation of the overall uncertainty; N is the number of samples; $x_i$ is the value of the i-th sample, and $\bar{x}$ is the mean of the sample.

T-tests have been performed to confirm the significance of the differences mentioned in the manuscript and to the differences are statistically significant. For the ± range that exists in the manuscript, such as "The HONO/$NO_2$ in the P2 period was higher than that in the P1 stage, especially in the daytime, although the values of HONO/$NO_2$ in both stages (P1: 0.036 ± 0.016; P2: 0.041 ± 0.038) were lower than that (0.052-0.080) reported by Cui et al (Cui et al., 2018). Subsequently, we further analyzed HONO$_{corr}$/$NO_2$ (details shown in Sect. 2.2). The HONO$_{corr}$/$NO_2$ attributed to secondary formation via heterogeneous reactions changed obviously after subtracting the interference of other HONO sources. As shown in Fig. S5, the daytime peak of HONO$_{corr}$/$NO_2$ in P2 became more prominent compared with that in Fig. 3e. At the same time, the HONO$_{corr}$/$NO_2$ (0.038 ± 0.035) in P1 was slightly lower than that in P2 (0.042 ± 0.034)" in the lines 471-479 of the manuscript, it's still a statistically significant change at 0.05 level according to the T-test result.

**Minor comments**

I've noted some more points below but the authors need to address the points noted above vs the overall approach, to have confidence that the results of their analysis allow them to draw the conclusions presented in the paper.

1. Introduction – reviews different sources for HONO and their contribution, but this mixes together very different environments (i.e. the relative importance of different sources will vary for the measurement site vs a road tunnel vs a bare soil location vs the marine boundary layer vs livestock). Suggest distilling this to assess the key factors at the measurement location, ie city center.

**Response:** Thank you for your good suggestion. In lines 67-135 in the revised manuscript, we revised the description as you suggested.

In the revised SI, we have added Table R2 as Table S1 and updated the sentence "The sources of atmospheric HONO consist of direct emissions and secondary

[revised manuscript text omitted]

**Table R2.** Summary of HONO observation sites and source contributions

| Location (References) | Date | Measurement area | Site situation | Source contribution |
|---|---|---|---|---|
| Antarctica (Bond et al., 2023) | 2022.01 | Research Station | Clean area, covered with ice and snow | Photolysis of nitrate in snow is very important, and its contribution is 10 times greater than the reaction between NO and OH. |
| Shenzhen (Tang et al., 2024) | 2019.10 | natural ecological area | Along the coast, there are fewer human activities and more vegetation. | Photolysis of large amounts of nitrate in coarse particles completely compensates for unknown sources during the daytime (66%). |
| China (Xing et al., 2023) | 2018.05 | sea edge | coastal | In inland areas, the $NO_2$ heterogeneous reaction on the ground is more important; in coastal and ocean cases, the contribution of aerosol surfaces is greater. |
| Idaho (Chai et al., 2021) | 2018.08 | wildfire zone | Smoke collected near five wildfires | In the aging smoke during the daytime, the heterogeneous conversion of $NO_2$ reaches 85%, followed by NO+OH. |
| Guangzhou (Li et al., 2012) | 2006.07 | rural area | Close to farmland, low traffic emissions | The main source at night is NO+OH and the heterogeneous conversion of $NO_2$ on the ground, and traffic can be ignored |
| Cyprus (Meusel et al., 2018) | 2016.04 | rural area | Along the coast, a lot of vegetation is exposed | Emissions from soil and biological soil crusts are important. |
| Melpitz (Ren et al., 2020) | 2018.04 | rural area | Nearby are meadows, agricultural areas, and forests | Nocturnal HONO: Heterogeneous conversion of ground $NO_2$ dominates, and traffic emission is a secondary source. |
| Wangdu (Liu et al., 2019b) | 2014.06 | rural area | Intensive agricultural activities and no traffic emissions | Noonday HONO: Soil emissions account for 80%. |
| Wangdu | 2017.12 | rural area | No traffic emissions, | Noonday HONO: The heterogeneous conversion of $NO_2$ on |

| | | | | |
|---|---|---|---|---|
| (Xue et al., 2020) | | | surrounded by farmland | the ground is 36%, NO+OH is 34%, and the others can be ignored. |
| Wangdu (Song et al., 2022) | 2020.06-2020.09 | rural area | No traffic emissions, surrounded by farmland | Noonday HONO: The heterogeneous conversion of $NO_2$ on the ground is dominant (43-62%), followed by NO+OH 12-38%, and the rest are less than 5%. |
| Wangdu (Zhang et al., 2023) | 2020.09-2021.08 | rural area | Seriously affected by agriculture and animal husbandry | Direct emissions from rural areas, including animal husbandry, account for 39-45% and cannot be ignored. |
| Hongkong (Zhang et al., 2016) | 2011.08 | suburbs | Near the airport, surrounded by vegetation and close to the South China Sea | The heterogeneous conversion of $NO_2$ on the ground is 42%, soil emission is 29%, marine source is 9%, NO+OH is 6%, aerosol surface conversion is 3%, and traffic is 2%. |
| Hongkong (Xu et al., 2015) | 2011.08-2012.05 | suburbs | Areas near airports and highways are mostly covered by vegetation. | Nocturnal HONO: Traffic dominates in the first half of the night (59%), and the heterogeneous conversion of $NO_2$ on the ground dominates in the second half of the night. |
| Heshan (Fu et al., 2019) | 2017.01 | suburbs | Lots of vegetation and farmland, with some scattered villages | Heterogeneous conversion of $NO_2$ is 72%, traffic is 8%, and NO+OH is 3%. Noonday HONO: Photolysis of nitrate accounts for more than 50%. |
| Taizhou (Ye et al., 2023) | 2018.06 | suburbs | Borders farmland and fish ponds | Noonday HONO: The heterogeneous conversion of $NO_2$ on the ground is 71%, followed by NO+OH, traffic, and aerosol surface conversion. Nocturnal HONO: Heterogeneous conversion of $NO_2$ on the ground is dominant (55%). |
| Beijing (Tong et al., 2015) | 2014.11 | urban area | Densely populated and busy with traffic | Nocturnal HONO: Traffic emission is 40%, NO+OH is 42%, and others are 18%. |
| | | suburbs | By the lake, with farmland | Nocturnal HONO: Traffic emission is 8%, NO+OH is 11%, |

| Location (Reference) | Date | Area | Description | Findings |
|---|---|---|---|---|
| | | | nearby | and others are 81%。 |
| Beijing (Tong et al., 2016) | 2014.12 | urban area | Densely populated and busy with traffic | Nocturnal HONO: Traffic emission is dominant (49%), and the reaction of NO and OH is also important. |
| | | suburbs | By the lake, with farmland nearby | Nocturnal HONO: Heterogeneous conversion of $NO_2$ is the main source, and traffic is 10%. |
| Beijing (Zhang et al., 2019a) | 2006.08 | urban area | Mixed residential, commercial, and transportation area | Nocturnal HONO: Traffic is 41%, ground heterogeneous conversion is 27%, and aerosol surface conversion is 20%. Noonday HONO: ground heterogeneous conversion is 66%, and aerosol surface conversion is 19%. |
| Beijing (Zhang et al., 2019c) | 2016.12 | urban area | Densely populated and busy with traffic | Nocturnal HONO: Traffic emission is dominant, reaching 52%, and heterogeneous conversion is not an important pathway. |
| Beijing (Meng et al., 2020) | 2016.12 | urban area | Mixed residential, commercial, and transportation area, 325m vertical observation. | High altitude during haze: HONO is dominated by heterogeneous conversion on the aerosol surface; Near the ground: Heterogeneous conversion of $NO_2$ on the ground is dominant, followed by traffic, accounting for 29% |
| Beijing (Gu et al., 2021) | 2017.05 2018.01 | urban area | Mixed residential, commercial, and transportation area | Noonday HONO: The light-induced heterogeneous transformation of $NO_2$ on the ground is dominant, and aerosol surface conversion can be ignored. Noonday HONO: NO+OH is dominant. |
| Beijing (Liu et al., 2020c) | 2018.02- 2018.07 | urban area | Mixed residential, commercial, and transportation area | Nocturnal HONO: Traffic emission is dominant, reaching 50%, and heterogeneous conversion is not an important pathway. Noonday HONO: Nitrate photolysis and NO+OH are important. |

| Location | Date | Area | Description | Findings |
|---|---|---|---|---|
| Beijing (Zhang et al., 2020) | 2018.04 | urban area | Mixed residential, commercial, and transportation area, 325m vertical observation. | At different altitudes, the heterogeneous conversion of $NO_2$ is the most important source, accounting for more than 70%. Among them, the aerosol surface is dominant. |
| Beijing (Jia et al., 2020) | 2018.08 | urban area | Mixed residential, commercial, and transportation area | Traffic is 18%, NO+OH is 31% (clean) and 7% (haze), and the aerosol surface conversion can reach up to 88%, which is very low on the ground. Nitrate photolysis is 15%, Noonday HONO: NO+OH is 22%, traffic is 19%, and |
| Beijing (Liu et al., 2021) | 2018.06 / 2018.12 | urban area | Mixed residential, commercial, and transportation area | Heterogeneous conversion on the aerosol surface is 19%。 Noonday HONO: Heterogeneous conversion on the aerosol surface is 30%, Heterogeneous conversion on the ground is 25%, and traffic is 20%. |
| Beijing (Zhang et al., 2022) | 2019.01 | urban area | Densely populated and busy with traffic | Traffic is 28%, Heterogeneous conversion on the ground is 27%, and aerosol surface conversion is 15%。 |
| Beijing (Li et al., 2021b) | 2019.06 | urban area | Mixed residential, commercial, and transportation area | Nocturnal HONO: The heterogeneous conversion of $NO_2$ is the main pathway, followed by NO+OH. Traffic is 30%. |
| Shijiazhuang (Liu et al., 2020b) | 2019.12-2020.03 | urban area | mixed traffic and residential area | Nocturnal HONO: The heterogeneous conversion of $NO_2$ on the ground is dominant, followed by aerosol surface conversion. |
| Beijing-Tianjin-Hebei (Zhang et al., 2019b) | 2017.12 | urban area | Less traffic emissions and intensive agricultural activities | Nocturnal HONO: Traffic and heterogeneous conversion of $NO_2$ are the main sources. |
| Xi'an | 2015.08 | urban area | Mixed residential, commercial, | Nocturnal HONO: The heterogeneous conversion of $NO_2$ is |

| | | | | |
|---|---|---|---|---|
| (Huang et al., 2017) | | | and transportation area | the main pathway, followed by NO+OH. Traffic is 19%. |
| Shanghai (Cui et al., 2018) | 2016.05 | urban area | Mixed residential, commercial, and transportation area | Nocturnal HONO: Heterogeneous conversion of $NO_2$ is the main source. |
| Nanjing (Zheng et al., 2020) | 2015.12 | urban area | To the west of the steel plant and petrochemical refinery, 15 kilometers from the city center | The heterogeneous conversion of $NO_2$ is dominant, accounting for 50%, and traffic is 11%. |
| Nanjing (Liu et al., 2019a) | 2017.11-2018.11 | urban area | Mixed residential, commercial, and transportation area | Traffic is 23%, heterogeneous conversion on the ground is 36%, Soil emissions can reach 40% in July and August. The aerosol surface conversion reaches 40% (severe haze periods). |
| Changzhou (Shi et al., 2020) | 2017.04 | urban area | Mainly residential and commercial areas, with no roads and industrial activities, | Nocturnal HONO: Heterogeneous conversion of $NO_2$ is 54%, traffic is 32%, and NO+OH is 14%. Noonday HONO: Nitrate photolysis is important. |
| Guangzhou (Yu et al., 2022) | 2018.10 | urban area | mixed traffic and residential area | Nocturnal HONO: The three main sources are the heterogeneous conversion of $NO_2$ on the ground, traffic, and NO+OH. The aerosol surface conversion and soil emissions are not important. |
| Birmingham (Kramer et al., 2020) | 2016.11 | urban area | road tunnel | Traffic is dominant, accounting for 66% (up to 86%), the heterogeneous conversion of $NO_2$ is only 5%, |

2. L174 – NO$_2$ measured by 42i – selectivity for NO$_2$, the NO$_2$ data will include other NO$_y$ species (including HONO).

**Response:** Thank you for your good comments. This is taken into account in our observations, so in the calculations, the NO$_2$ concentration has been subtracted from the HONO concentration. In lines 174-175 in the revised manuscript, we added a sentence "Notably, the NO$_2$ measured by 42i includes HONO. Thus, it has been corrected".

3. L257 how was j$_{(HONO)}$ calculated

**Response:** Thank you for your comments. j$_{(HONO)}$ is simulated in MCM using j$_{(NO2)}$ data observed at our site. The details are as follows: In each case, variation of photolysis rate with solar zenith angle can be described well by an expression of the following form,

$$J = l\ (\cos \chi)^m \exp(-n.\sec \chi)$$

by optimizing the values of the three parameters, $l$, $m$, and $n$ are the parameters of the reaction respectively, $\chi$ is *the* solar zenith angle (Saunders et al., 2003). In line 257 in the revised manuscript, we added a sentence "$J_{HONO}$ is simulated in a box model using $J_{NO2}$ data observed at our site".

4. L190 heterogeneous reactions of NO$_2$ have been accounted for – I didn't follow this section, may need rewording.

**Response:** Thank you for your suggestion. I am sorry for confusing you. In the revised manuscript, we made changes and deleted the "have been accounted for" in "The sources including vehicle emissions ($E_{vehicle}$), soil emissions ($E_{soil}$), the reaction of NO and OH ($P_{NO-OH}$), the photolysis of particulate nitrate ($P_{nitrate}$), and the heterogeneous reaction of NO$_2$ ($P_{aerosol}$ and $P_{ground}$)" in lines 190-192.

5. L218 what is the conversion factor alpha?

**Response:** Thank you for your good suggestion. It is a unit conversion factor of emission flux from g m$^{-3}$ s$^{-1}$ to ppb h$^{-1}$. In lines 218-219 in the revised manuscript, we update the "conversion factor ($\alpha$, from g m$^{-3}$ s$^{-1}$ to ppb h$^{-1}$)".

6. L227 using NO$_2$ and CO – NO$_2$ concentrations will vary with a lifetime of a minute

or so (with respect to the NOx-O$_3$ PSS) and 6-12 hours our so (with respect to NOx removal). CO concentrations will vary with a lifetime of several weeks. Is it valid to use both in the same in situ emission analysis?

**Response:** Thank you for your good suggestion. When calculating $k_{het}$, equation 3 does not consider the impact of the boundary layer on the concentration of pollutants. CO is considered to be a relatively stable substance and is usually used for partially eliminating the variation of the boundary layer height. Thus, equation 4 corrects the boundary layer variation by normalizing to CO concentration. This method has been widely used to calculate the heterogeneous uptake kinetic of NO$_2$ on aerosol in previous studies (Zhang et al., 2020; Zhang et al., 2019c; Li et al., 2012). It is expected to get a more accurate $k_{het}$ because the influence of the variation of boundary layer height has been accounted for. It is worth noting that during a year of HONO observations in coastal cities in southeastern China (Hu et al., 2022), the author compared the results of balancing NO$_2$, CO, and BC and found that CO performed well, and finally selected CO as the parameter for $k_{het}$ calculation.

In lines 233-235 in the revised manuscript, we have pointed out that "To decrease the contribution of boundary layer height variation on the $k_{het}$ calculations, we normalized HONO concentration to CO concentration as the same as reported in the literature (Zhang et al., 2019c; Li et al., 2012)".

7. L257 / table 1 – what is the estimated OH concentration & how does it compare with measurements – the values in L436 (presumably 24 hour mean, around 4-7e5) seem much lower than those observed in wintertime Beijing (2.7e6 as 24-hour mean; Slater et al., 2020).

**Response:** Thank you for your good suggestion. The article by Slater et al. in 2020 describes it as follows: "Averaged over the full observation period, the mean daytime peak in radicals was $2.7\times10^6$, $0.39\times10^8$, and $0.88\times 10^8$ cm$^{-3}$ for OH, HO$_2$, and total RO$_2$, respectively". Our estimated 24-hour mean concentration of OH ranges from $1.1\times10^5$ to $2.5\times10^6$, which is very close to the daytime peak observed in Beijing in winter ($2.7\times10^6$) (Slater et al., 2020). The daily mean value ($4-7\times10^5$) is therefore lower than the maximum value at noon. As shown in Figure R4 or S3, the estimated OH concentrations are in good agreement with the observations in Beijing.

[Figure]

**Figure R4.** Diurnal variation of OH concentrations observed in different areas of the North China Plain (a-d) (Tan et al., 2017; Tan et al., 2018; Ma et al., 2019; Tan et al., 2020) and parameterized fitting in this study (e).

8. L270 Parameterisation of OH concentrations. We might expect that the main source of OH in Beijing is $HO_2 + NO$, the main sink for OH is $OH + NO$, and the main primary source of OH is HONO photolysis (one can then argue about if HONO is acting as a primary source or a reservoir). The parameterization given was developed for rural sites (as the authors note), where NO levels would be much lower, and was developed prior to more recent understanding of HONO abundance (it is over 20 years old). Is it valid to use? L286 – considering these above I do not agree that we can be optimistic about the estimated OH concentrations.

**Response:** Thank you for your good suggestion. Notably, this parameterization scheme was developed based on measurements at rural sites (Ehhalt and Rohrer, 2000), where NOx concentrations were lower than in urban environments. Alicke et al. (Alicke, 2002) found OH concentrations estimated with this scheme were in good agreement with those calculated according to a pseudo-steady state method during the pollution period in urban environments (such as Milan) although some uncertainty was expected.

In our previous study (Liu et al., 2020d), we also found that the estimated OH concentrations using this method were comparable with those observed values in the North China Plain (Tan et al., 2019). Thus, daytime OH concentrations estimated using this method should be overall credible although the uncertainty is inevitable. Fig R2 (Fig S2) summarizes the observed OH concentrations in the North China Plain. The results estimated in this study are slightly lower than those observed in Wangdu, but almost consistent with those in Beijing and Huairou. In summary, although the parameterization was developed many years ago, judging from the performance results, it still has excellent performance in fitting OH.

In lines 284-287 in the revised manuscript, we have highlighted the points "The results estimated in this study are slightly lower than those observed in Wangdu (Rural), but almost consistent with those in Beijing (Urban) and Huairou (Suburb). In summary, we should be optimistic about the estimation of OH concentration."

9. L370 – $PM_{2.5}$ components increased obviously in P2 vs P1 – from the plot it is not obvious to me that they do: is there a statistically significant change? What about changes in the meteorology?

**Response:** Thank you for your comment. The increasing trend in Figure 1 is not very obvious, and we further summarize their concentrations in P1 and P2 in Table S4. In the revised manuscript, we added the sentence "It can be seen from Figure 1 and Table S4 in SI. All the major components of $PM_{2.5}$, including sulfate, nitrate, ammonium, chloride, and organic aerosol, increased obviously in P2 compared to P1." in lines 370-372. The difference in $PM_{2.5}$ between the two periods is statistically significant (P<0.05). In the revised manuscript, we added the sentence "The $PM_{2.5}$ concentration after deweather increased significantly from 45.22±28.56 in P1 to 67.92±57.97 µg m$^{-3}$ in P2 at a confidence level of 0.05, with an increase of 50.2%." in lines 353-355.

10. L410 the traffic index data are useful. Consider showing P1/P2 on these plots. Consider changing the box/whisker plot to linear (not log) – this will assist the reader to follow which differences are statistically significant.

**Response:** Thank you for your good comment. We have changed the logarithmic plot of Figure 2b into a linear plot.

[Figure]

**Figure R2. (a)** Times series of HONO, traffic index, and HONO/NO$_2$, **(b)** Box plots of HONO, HONO/NO$_2$, and the traffic index in Beijing during different periods (BCNY=P1, LOCK=P2).

11. concentrations of NO changed: You cannot conclude this without a statistical test, esp for NO$_2$ and HONO.

**Response:** Thanks for your suggestion. Regarding the decrease in NO concentration, we did a T-test and found that there was a significant difference in the concentrations between the two time periods (P<0.05). In lines 588-589, it has been pointed out as "This can be explained by the significant decrease (P < 0.05) in NO and direct emissions of HONO from traffic."

12. L449 Figure 3 the shift in diurnal profiles is interesting, explore further (esp panel 5, HONO/NO$_2$)?

**Response:** Thanks for your suggestion. The daily variation curves can provide us with a lot of information. For example, HONO, NOx, SO$_2$, and O$_3$ show similar diurnal variations, while the concentrations of HONO, NOx, and SO$_2$ decrease obviously during the COVID lockdown when compared to that before the COVID lockdown and O$_3$ concentrations increase. However, the diurnal curves of HONO/NO$_2$ and PM$_{2.5}$×NO$_2$ show obvious daytime peaks during the COVID lockdown, which are different from those before the lockdown. In addition, their nocturnal values are comparable between the two periods, while their daytime values are higher during the COVID lockdown than those before the lockdown. Because both HONO/NO$_2$ and PM$_{2.5}$×NO$_2$ are indicators for the heterogeneous reaction of NO$_2$ on aerosols to form HONO, these results imply that the heterogeneous reaction rate during the COVID lockdown should

be higher than that before the lockdown. However, the absolute concentrations of HONO during the COVID lockdown are lower than those before the lockdown. It suggests that heterogeneous reactions should be an unimportant source of HONO in Beijing.

In lines 485-487 in the revised manuscript, we emphasize this point "In summary, we propose that during our observation period, heterogeneous reactions of $NO_2$ should have a relatively minor contribution to HONO production".

13. L479 contribution (not interference) of other HONO sources. This matters vs literature discussion of interferences in some HONO measurement approaches.

**Response:** Thanks for your good suggestion. In our $HONO_{corr}$ calculation method, primary and other secondary sources of HONO not attributing to heterogeneous reactions have been subtracted. Thus, the obtained value is more reasonable to present a heterogeneous conversion of $NO_2$.

To make it clearer, we revised the sentence in lines 478-479 in the revised manuscript as "The $HONO_{corr}$/$NO_2$ attributed to secondary formation via heterogeneous reactions changed obviously after subtracting other secondary HONO sources".

14. L403 ifs there any info on changes in fleet composition? Big change in total traffic, but the greatest change may be in discretionary journeys (private cars) while deliveries etc(potentially much greater per-vehicle emitters) may have continued.

**Response:** Thanks for your suggestion. Unfortunately, changes in fleet composition are unavailable. The situation you mentioned may exist, that is, some delivery trucks still have continued. It should be noted that if the HONO emissions by trucks along with the heterogeneous reaction sources increase during the lockdown, an increase in HONO concentrations should be observed. Thus, we ascribe the reduction of HONO concentrations to the decrease in total emissions of HONO from vehicles as supported by the decreases in the traffic index. Therefore, even if the emissions of trucks increase due to the load, this increase is still insignificant compared to the overall emissions of vehicles before the lockdown.

15. L612 the agreement is good to see but is there really confidence in the combined uncertainty of the terms entering the calculation – especially [OH] – to have confidence? L625 same point.

**Response:** Thanks for your good suggestion. Your opinion is very reasonable. Although we have calibrated each parameter strictly and as accurately as possible, we still cannot guarantee 100% accuracy. For example, for the fitting of OH radicals, although our fitting agrees well with the observations in Beijing, this still does not mean that our results are 100% correct. Considering the complex sources of HONO, uncertainty in the calculation is inevitable, while it is acceptable. In particular, the estimated hourly HONO concentrations are well in agreement with the observed ones. There are 27.2% in total. This means that the simultaneous equations (6 sources and 4 sinks) have been verified 10,000 times. To our best acknowledgment, this is the first time to constrain the parameterization for HONO source budget analysis based on steady-state analysis.

16. L638 plus – these changes must be considered in the context of (potential) changes in meteorology between the two periods – it can be very misleading to simply compare means calculated from two different, fairly short, date periods.

**Response:** Thanks for your good suggestion. Your suggestion is great. Changes in atmospheric pollutant concentration are affected by emissions and meteorology. The concentrations and relative changes of each pollutant after deweather are recorded in Table R2 (Table S6).

In the revised manuscript, we added the sentence "After removing meteorological factors, the change proportions of $PM_{2.5}$ concentration in the two stages were -4.26% and -2.28% respectively. The HONO changes were -8.25% and -3.77% respectively, the CO changes were -9.27% and -6.17% respectively, and the $SO_2$ changes were +8.61% and +0.68% respectively. The change proportions are all less than 10%, which means that the impact of changes in meteorological factors on $PM_{2.5}$, HONO, CO, and $SO_2$ is very weak. However, the change proportions of NO in the two stages were -16.18% and +32.79%, respectively, and $O_3$ was +39.64% and +6.15% respectively. The change ratio is greater than 30%, indicating that NO and $O_3$ are greatly affected by meteorology. In addition, the changes in $NO_2$ were -13.75% and -4.81% respectively, implying that $NO_2$ is also affected by meteorological factors. From the entire observation period, except for $O_3$, the changes of other species in the two periods fluctuated between 2.3% and 7.8% after deweather, all less than 8%. In general, after removing the meteorological effects, NO increased by 79%, $NO_2$ increased by approximately 29%, HONO decreased by approximately 43%, and $PM_{2.5}$ increased by approximately 50%. It is worth noting

that $O_3$ increased by about 33%, which is much lower than the change in observed values (75.08%) (as shown in Table S6)." lines 639-653.

[revised manuscript text omitted]

---

## Referee Report (RR1)

I am grateful to the authors for the extensive and detailed work to address the comments raised previously – these substantially improve the manuscript. In particular, the application of deweathering approaches to separate the meteorological changes from emission changes in assessing the pre/post lockdown concentrations significantly increases the robustness of the inference regarding changes in HONO due to emissons.

I am still slightly unclear if the (change in) deweathered concentrations are used in the subsequent analysis, or the raw concentrations – with implications for the emissions sources following on – maybe needs to be clarified explicitly.

The source function mapping for HONO (R2) is useful, it would be valuable to add to the manuscript conclusions the caveat that changes in airmass – as indicated on this figure (would be better to add the measurement site IAP also) exist between the two sampling periods, and this is a limitation of the analysis. I'm not sure how much can be drawn from the observation that there is an r of 0.79 and gradient of 0.6 between to separate measurement locations. These points – and the wider dependence on a large number of parameters, all inevitably with some uncertainty – do have consequences for the accuracy / precision of the final values presented, but I am comfortable with the language used in the abstract – the authors could consider reflecting this point at the start of the conclusions also.

I'd ask the authors to consider how the statistical difference is presented – reviewing Figure S5 the (HONOcorr/NO2) ratios / diurnal patterns clearly differ, but presenting these as the mean +/- SD values [ (0.038 +/- 0.035), compared to a value of (0.042 +/- 0.034) ] doesn't convey this to the audience nearly as well as the figure does – maybe include (the figure) in the main manuscript ?

Minor point -jNO2 if derived from the MSM / Saunders et al approach doesn't account for clouds etc (it's a clear sky parameterisation), but maybe this was normalised to measured jNO2

---

## Author Response (AR2)

Dear Editor,

We appreciate your careful consideration of our manuscript. We have carefully responded to all comments from you and the reviewer **point-by-point** and have revised the manuscript accordingly. These revisions are described in detail below.

**Editor's comments**

The revised manuscript was reviewed again by Referee #2, who is now largely satisfied with the answers and revisions. Before the manuscript can be published, some more clarifications are needed as suggested by the reviewer.

 **Response:** Thank you for your positive comments and good suggestions. We will respond to your comments point-by-point below.

1. In particular, the HONO source contributions in the abstract should be provided with error bars (e.g., nighttime vehicle emissions of 53% +/- ?), and in the conclusions, it should be mentioned which of the assumed parameters in the budget analysis cause the largest uncertainties in the estimated HONO source and how much.

 **Response:** Thank you for your good suggestions. In the revised manuscript, we have added error bars for contributions from different sources of HONO. Through sensitivity analysis, we found that the source of HONO is most sensitive to OH radicals and $J_{NO_3^-}$. The uncertainty caused by OH radicals is ±5% using the observed values reported in the literature as constraints. For $J_{NO_3^-}$, there is a lack of direct observational data. Although parameter optimization was carried out in this study in combination with the results of the literature, it still brought the greatest uncertainty to the HONO source and sink assessment, with a value of ±19%.

 In lines 694-695 in the revised manuscript, we added a sentence "Through uncertainty assessment, it was found that the assumption of $J_{NO_3^-}$ would have the greatest uncertainty, with a standard deviation of ±19%. Nevertheless, this study confirms that reducing anthropogenic emissions can indeed reduce the concentration of HONO in the atmosphere."

2. In Table S3, a definition of the 'Sensitivity' (last column) is missing.

 **Response:** Thank you for your good suggestions. In Table S3 in the revised SI, we

added the sentence "The source of HONO is affected by many factors, and its concentration varies with any one of these factors. The sensitivity here is calculated by univariate analysis, that is, observing the changes in HONO concentration by changing only one variable but with all other variables unchanged,"

3. Some numbers in the conclusions sections are provided with unreasonable precisions (e.g. " HONO changes were -8.25% and -3.77%"). Please correct.

**Response:** Thank you for your good suggestions. In the revised manuscript, we have modified the significant figures of the data. For example, "HONO changes were -8.25% and -3.77%" was changed to "HONO changes were -8.3% and -3.8%".

**Reviewer #2**

I am grateful to the authors for the extensive and detailed work to address the comments raised previously – these substantially improve the manuscript. In particular, the application of deweathering approaches to separate the meteorological changes from emission changes in assessing the pre/post lockdown concentrations significantly increases the robustness of the inference regarding changes in HONO due to emissions.

**Response:** Thank you for your positive comments and good suggestions. We will respond to your comments point-by-point below.

1. I am still slightly unclear if the (change in) deweathered concentrations are used in the subsequent analysis, or the raw concentrations – with implications for the emissions sources following on – maybe needs to be clarified explicitly.

**Response:** Thank you for your good suggestions. We conducted the budget analysis of HONO using the deweathered concentrations as shown in Figure R1 (or Figure S10). The corresponding results are shown in Figure S9 (Figure R2) using the raw observed concentrations. The daytime source contributions of HONO almost did not change in both cases (using raw data or deweathered data) whatever the periods. However, the nocturnal contribution of vehicle emissions increased from 53% with the raw dataset to 63% with the deweathered dataset, along with a decrease of heterogenous reactions on ground surface from 31% to 19% before the Chinese New Year. It also slightly increased from 40% to 45%

for vehicle emissions, accompanied by a decrease of heterogenous reactions on ground surface from 47% to 42% during the COVID-19 lockdown. This further highlights the importance of vehicle emissions to nocturnal HONO sources in Beijing.

In lines 628-640 in the revised manuscript, we added a sentence "To explore whether meteorological factors have an impact on the sources of HONO, we conducted the budget analysis of HONO using the deweathered pollutant concentrations. The results are shown in Fig. S10. When compared with the sources of HONO calculated using the raw concentration dataset (Fig. S9), it can be seen that deweathering has little effect on the daytime sources of HONO. For the nighttime source of HONO, however, deweathering caused the proportion of traffic emissions during BCNY increasing from 53% to 63% before the CNY or from 40% to 45% during the COVID-19 lockdown. The contribution of heterogeneous reactions of $NO_2$ on ground surfaces decreased from 31% to 19% before the CNY or from 47% to 42% during the COVID-19 lockdown. These results further highlight the importance of vehicle emissions to nocturnal HONO sources in Beijing.

Therefore, regardless of whether the impact of meteorological conditions on the source of HONO is considered, we can conclude that traffic-related emissions, rather than heterogeneous reactions of $NO_2$ were the main HONO source at night in Beijing in the typical emission patterns of air pollutants."

[Figure]

Figure R1. The percentage of daytime and nighttime contributions from different sources in (a,c) BCNY and (b,d) COVID. Pollutant concentrations are all de-weathering concentrations.

[Figure]

Figure R2. The percentage of daytime and nighttime contributions from different sources in (a,c) BCNY and (b,d) COVID. Pollutant concentrations are all raw concentrations.

2. The source function mapping for HONO (R2) is useful, it would be valuable to add to the manuscript conclusions the caveat that changes in airmass – as indicated on this figure (would be better to add the measurement site IAP also) exist between the two sampling periods, and this is a limitation of the analysis. I'm not sure how much can be drawn from the observation that there is an r of 0.79 and gradient of 0.6 between two separate measurement locations. These points – and the wider dependence on a large number of parameters, all inevitably with some uncertainty – do have consequences for the accuracy/precision of the final values presented, but I am comfortable with the language used in the abstract – the authors could consider reflecting this point at the start of the conclusions also.

   **Response:** Thank you for your good suggestions. We conducted a potential source contribution function of HONO for the two observation sites, as shown in Figure R3 (or Figure S2) found that they are highly consistent. However, this still cannot rule out the influence of meteorological changes. In lines 683-693 in the revised manuscript, we added a sentence "We conducted a potential source contribution function (PSCF, Fig S2) analysis in different periods, i.e., BCNY and COVID, at the BUCT station and further compared the PSCF of HONO at BUCT station with that at the Institute Atmospheric Physics (IAP) station, which is around 8 km from BUCT station, from January 24, 2022, to January 31, 2022, when the data were available. The PSCF patterns were highly similar in different periods or locations. These results mean that the air mass should be consistent during the COVID-19 lockdown and BCNY and HONO should be evenly distributed in Beijing. Thus, the impact of meteorological changes on the accuracy of observations cannot be ruled out, which is also a limitation of this study, but its influence should be comparable between BCNY and the COVID lockdown. And the conclusions drawn based on the observations at BUCT should represent the situation in Beijing". And in lines 650-651 in the revised manuscript, we added a sentence "It is worth noting that in addition to primary emissions, meteorological changes will also affect changes in atmospheric pollutant concentrations". And in lines 666-668 in the revised manuscript, we also added a sentence "Although we have tried to assess the impact of meteorological factors quantitatively, this still carries some uncertainty. In particular, uncertainty is inevitable for the source assessment of

substances such as HONO that are affected by a large number of parameters".

[Figure]

**Figure R3.** The potential source contribution function (PSCF) maps for the concentration of HONO (a and b are BCNY and COVID; c and d are BUCT and IAP stations, respectively). The comparison period of c and d is 2022.01.24-2022.01.31, and the trajectory of the air mass is 12 hours.

3. I'd ask the authors to consider how the statistical difference is presented – reviewing Figure S5 the ($HONO_{corr}/NO_2$) ratios / diurnal patterns clearly differ, but presenting these as the mean +/- SD values [(0.038 +/- 0.035), compared to a value of (0.042 +/-0.034)] doesn't convey this to the audience nearly as well as the figure does – maybe include (the figure) in the main manuscript?

**Response:** Thank you for your good suggestions. When we present the values in Figure S5 as the average values for the whole day, the data cannot reflect the significant difference between the two periods. Therefore, we describe the average of the daytime with significant differences here to better highlight their statistical differences. In the revised manuscript in lines 478-480, we have modified it to "As shown in Fig. S5, the daytime peak of $HONO_{corr}/NO_2$ in P2 became more prominent compared with that in Fig. 3e, while the daytime (8:00 - 18:00) $HONO_{corr}/NO_2$ ($0.022 \pm 0.014$) in P1 was significantly ($P < 0.05$) lower than that in P2 ($0.040 \pm 0.053$)".

4. Minor point -$j_{NO2}$ if derived from the MSM/Saunders et al approach doesn't account for clouds etc (it's a clear sky parameterization), but maybe this was normalised to measured $j_{NO2}$

**Response:** Thank you for your suggestions. Our $J_{NO2}$ measurement equipment is a modified version of the original design by Junkermann et al., 1989 (Junkermann and Platt, 1989). The measuring principle is that the variation of the photolysis rate of $NO_2$ depends almost exclusively on the photochemical flux in the wavelength interval 300 to 420 nm. Photoelectric measurement of the photochemical flux in this spectral range gives a good approximation to the photolysis of $NO_2$ [$J_{(NO2)}$]. The instrument consists of two detectors each covering one hemisphere with a nearly uniform angular response. The two detectors are mounted in opposite directions, one facing up towards the sky and one facing down. They consist of a set of quartz-domes, a set of optical filters, a radiation sensitive detector and an electronic board to provide a voltage output signal. Since the $J_{(NO2)}$ photometer measures the radiant flux rather than directly measuring the photodecomposition rate of $NO_2$, the instrument was calibrated to obtain standardized $J_{NO2}$ according to the photometer standards.

References:

JUNKERMANN, W. and PLATT, U.: A Photoelectric Detector for the Measurement of Photolysis Frequencies of Ozone and Other Atmospheric Molecules, Journal of Atmospheric Chemistry, 8, 203-227, 1989.

---

## Author Response (AR3)

Dear Editor,

We appreciate your careful consideration of our manuscript. We have carefully responded to your comment point-by-point and revised the manuscript accordingly. These revisions are described in detail below.

**Editor's comments**

Thank you for the clarifications in the revised version of your manuscript. There is one point that I had overlooked and would like you to check. In the Supplement, you specify the accuracy of your $j_{NO2}$ measurements as 1%. Such a good accuracy is hard to believe (other groups specify accuracies between 9% and 20%; e.g., Shetter et al., J. Geophys. Res. Vol. 108, D16, 8544, doi:10.1029/2002JD002932, 2003; Zou et al., Front. Environ. Sci. Eng. 2016, 10(6): 13). Please explain the accuracy of your $j_{NO2}$ data and provide a corresponding reference.

**Response:** Thank you for your comment. We are sorry for making such a mistake about the accuracy of $j_{NO2}$ measurements. We have wrongly reported the precision as the accuracy of our instrument. The accuracy should be 11% (Shetter et al., 2003). In the revised SI, we have corrected it and added a citation.

We calculated the precision of the $J_{NO2}$ according to the instrument's standard operating manual, which states that $J_{NO2}$ remains fairly constant with an accuracy of about 1% when the calibration factors of the two detectors do not change. The details are shown in Table R1.

Table R1. Some typical data of calibration check.

| Data Acquisition | Serial numbers | 1. Signal (V) | 2. Signal (V) | 3. Signal (V) | 4. Signal (V) | Total J(NO2) |
|---|---|---|---|---|---|---|
| Step1 Original position | 249/248 | 3.210/ 0.522 | 3.195/ 0.521 | 3.207/ 0.522 | 3.204/ 0.522 | $6.41 \times 10^{-3}$ |
| Step 2 Inverted | 248/249 | 0.463/ 3.620 | 0.461/ 3.625 | 0.460/ 3.622 | 0.461/ 3.622 | $6.33 \times 10^{-3}$ |
| Step 3 Inverted cleaned | 248/249 | 0.465/ 3.623 | 0.465/ 3.625 | 0.466/ 3.627 | 0.465/ 3.625 | $6.35 \times 10^{-3}$ |
| Step 4 Original cleaned | 249/248 | 3.230/ 0.527 | 3.238/ 0.527 | 3.225/ 0.522 | 3.231/ 0.525 | $6.36 \times 10^{-3}$ |

Note: 248 and 249 are the serial numbers of the two detectors. The photometer is mounted so that detector 249 faces up. Calibration Factor (249) = $1.72 \times 10^{-6}$ s$^{-1}$/mV; Calibration Factor (248) = $1.53 \times 10^{-6}$ s$^{-1}$/mV; Signal Ratio : (249)/(248) = 0.890.

References:

Shetter, R. E., Junkermann, W., Swartz, W. H., Frost, G. J., Crawford, J. H., Lefer, B. L., Barrick, J. D., Hall, S. R., Hofzumahaus, A., Bais, A., Calvert, J. G., Cantrell, C. A., Madronich, S., Müller, M., Kraus, A., Monks, P. S., Edwards, G. D., McKenzie, R., Johnston, P., Schmitt, R., Griffioen, E., Krol, M., Kylling, A., Dickerson, R. R., Lloyd, S. A., Martin, T., Gardiner, B., Mayer, B., Pfister, G., Röth, E. P., Koepke, P., Ruggaber, A., Schwander, H., and van Weele, M.: Photolysis frequency of NO2: Measurement and modeling during the International Photolysis Frequency Measurement and Modeling Intercomparison (IPMMI), Journal of Geophysical Research: Atmospheres, 108, 10.1029/2002jd002932, 2003.